# A latent class analysis of the socio-demographic factors and associations with mental and behavioral disorders among Australian children and adolescents

**Nahida Afroz**[1,2]*, **Enamul Kabir**[2], **Khorshed Alam**[3]

**1** Department of Statistics, Faculty of Science, Comilla University, Cumilla, Bangladesh, **2** School of Mathematics, Physics, and Computing, Faculty of Health, Engineering and Sciences, University of Southern Queensland, Toowoomba, Queensland, Australia, **3** School of Business, Faculty of Business, Education, Law & Arts, and Centre for Health Research, University of Southern Queensland, Toowoomba, Queensland, Australia

* n.afroz@cou.ac.bd, Nahida.Afroz@usq.edu.au

**Data Availability Statement:** In accessing this data should contact the Young Minds Matter: The second Australian Child and Adolescent Survey of Mental Health and Wellbeing Dataverse through

## Abstract

### Background

Previous studies have shown a relationship between socio-demographic variables and the mental health of children and adolescents. However, no research has been found on a model-based cluster analysis of socio-demographic characteristics with mental health. This study aimed to identify the cluster of the items representing the socio-demographic characteristics of Australian children and adolescents aged 11–17 years by using latent class analysis (LCA) and examining the associations with their mental health.

### Methods

Children and adolescents aged 11–17 years (n = 3152) were considered from the 2013–2014 Young Minds Matter: The Second Australian Child and Adolescent Survey of Mental Health and Wellbeing. LCA was performed based on relevant socio-demographic factors from three levels. Due to the high prevalence of mental and behavioral disorders, the generalized linear model with log-link binomial family (log-binomial regression model) was used to examine the associations between identified classes, and the mental and behavioral disorders of children and adolescents.

### Results

This study identified five classes based on various model selection criteria. Classes 1 and 4 presented the vulnerable class carrying the characteristics of "lowest socio-economic status and non-intact family structure" and "good socio-economic status and non-intact family structure" respectively. By contrast, class 5 indicated the most privileged class carrying the characteristics of "highest socio-economic status and intact family structure". Results from the log-binomial regression model (unadjusted and adjusted models) showed that children

email or lodge an online application from the following web link: http://www.youngmindsmatter.org.au/information/for-researchers/.

**Funding:** The author(s) received no specific funding for this work.

**Competing interests:** The authors have declared that no competing interests exist.

and adolescents belonging to classes 1 and 4 were about 1.60 and 1.35 times more prevalent to be suffering from mental and behavioral disorders compared to their class 5 counterparts (95% CI of PR [prevalence ratio]: 1.41–1.82 for class 1; 95% CI of PR [prevalence ratio]: 1.16–1.57 for class 4). Although children and adolescents from class 4 belong to a socio-economically advantaged group and shared the lowest class membership (only 12.7%), the class had a greater prevalence (44.1%) of mental and behavioral disorders than did class 2 ("worst education and occupational attainment and intact family structure") (35.2%) and class 3 ("average socio-economic status and intact family structure") (32.9%).

## Conclusions

Among the five latent classes, children and adolescents from classes 1 and 4 have a higher risk of developing mental and behavioral disorders. The findings suggest that health promotion and prevention as well as combating poverty are needed to improve mental health in particular among children and adolescents living in non-intact families and in families with a low socio-economic status.

## Background

Globally, it is estimated that 14% of 10-19-year-olds suffer from mental health difficulties, but they are commonly ignored and untreated [1]. For people, their families, and society as a whole, mental illness has a wide range of short and long-term negative consequences [2]. The numerous physical, psychological, and behavioral changes that adolescents undergo can lead to psychosocial and mental health issues, which can have major effects on their growth, productivity, and quality of life [3]. Furthermore, mental illnesses that arise throughout childhood and adolescence are well-known risk factors for mental disorders later in life [4–6]. Mental disorders among adolescents have been identified as one of the major public health concerns worldwide [1]. According to the Mission Australia's latest *Youth Survey Report 2022*, 33.9% of young people viewed mental health as a significant national concern. Almost three in 10 (28.8%) young people reported experiencing severe psychological distress, and nearly one-quarter (23.5%) stated feeling lonely most of the time. Little more than half of individuals (53.4%) have at some stage in their lifetimes required mental health care. In 2020, 49.9% of young people were optimistic about their future, but this proportion has steadily decreased since then [7]. Similar to many other developed countries, Australia experiences a large health burden due to mental disorders [8, 9]. Both incidence and prevalence of mental health conditions are rising significantly among young Australians. According to Irteja et al. (2020), 12-months prevalence rate of mental disorders among adolescents aged 13–17 years old was 34.7% [10]. Therefore, policymakers must be aware of the particular population group suffering from mental health at risk of developing such conditions so that appropriate, timely interventions can be facilitated. As a result, it is necessary to determine various classes based on relevant socio-demographic factors by using appropriate statistical methods, so that distinctive health policies can be applied according to the needs of each class. From the health policy framing and implementation perspectives, it is also important to examine the relationship between obtained classes and the mental health status of children and adolescents.

Various socio-demographic characteristics such as age, regional status, gender, parental education, parental employment status, and household income can predict mental health and aggressive behaviors in children and adolescents [11–13]. For instance, household income and

parental education had a greater impact on children's and adolescents' mental health issues than did parental unemployment or poor occupational status, which refers a low position in the occupational hierarchy [14]. Moreover, children of parents with university degrees are more likely to have greater levels of psychological well-being than children of parents without a university degree [15]. The mental health of children and adolescents is also expressively influenced by their family structure [16]. The family structure is a representation of the connection between one's living arrangements, marital status, and biological relatedness [17]. According to the Australian Bureau of Statistics (ABS) report (2013) on Australian families with children and adolescents, the majority of children (63–78%) in all age groups lived with both of their biological parents, 15–25% lived in a single-parent family, only 5–6% lived in a blended family, and 2–7% lived in a stepfamily [18]. In addition, 71% of all children lived with both natural parents (bio-logical parents), 4% were in stepfamilies, and 5% were in blended families [18]. The ABS report also stated that the percentage of adolescents living in their household varies with their age. Naturally, children under the age of 5 were more likely to live with both their biological parents (78%), followed by those 5–9 years in age (72%). The least likely to be in this scenario (63%) were those 15–17 years old [18]. Adolescents from non-intact families had a lower perception of family functioning than did adolescents from intact families [19]. Numerous studies [20, 21] have shown that children from non-intact households are more likely to experience negative psychological consequences than children from intact families. An Indonesian study found that the mental health problems of adolescents were associated with their educational level and area of residence [22]. A systematic review revealed that children and adolescents from low-income families were two to three times more likely to suffer mental health issues [14]. To date, in all these studies, those socioeconomic and demographic variables were only analyzed separately not in combination. According to Bronfenbrenner's ecological systems theory, child development is a complex system of relationships affected by multiple levels of the surrounding environment, from the immediate settings of the family to the broad environment which includes society and culture [23]. Therefore, instead of analyzing each socio-demographic factor separately, in the present study, we considered a latent class analysis (LCA) method, which allows for a person-centered approach to understanding how a wide range of socio-demographic factors may cluster together to reflect distinct social classes within a large and heterogeneous population.

The literature describes diverse classifications of the cluster analysis of influencing factors of health status. A US study identified classes according to varying combinations of health lifestyle behaviors using LCA and examined the associations between these classes and suicidal behaviors among adolescents [24]. Similarly, Hoogstoel et al. used LCA to identify different profiles from a list of risky behaviors among adolescents in Mauritius and then determined the associations of suicidality among adolescents with these profiles [25]. Daundasekara et al. (2022) likewise used LCA to identify socio-economic and health risk profiles among mothers of young children to predict longitudinal risk of food insecurity [26]. An Australian National University (ANU) report revealed a greater stratification in Australian society. Using LCA, the report showed that Australian society is best explained by five classes based on the economic, social and cultural capital of respondents [27]. The Main variables of interest of this report were the socio-demographic factors of the adolescents and their corresponding households. Depending on income, regional status, employment, family history, and other variables, every country and its people are categorized into numerous social classes or levels. While class distinctions are not completely evident or defined by any formal organization, they do exist in terms of how people live, conduct themselves, and spend money [27]. There are five main social classes in Australia [27]. Fig 1 represents the Australian social hierarchy with the mean characteristics of the different classes.

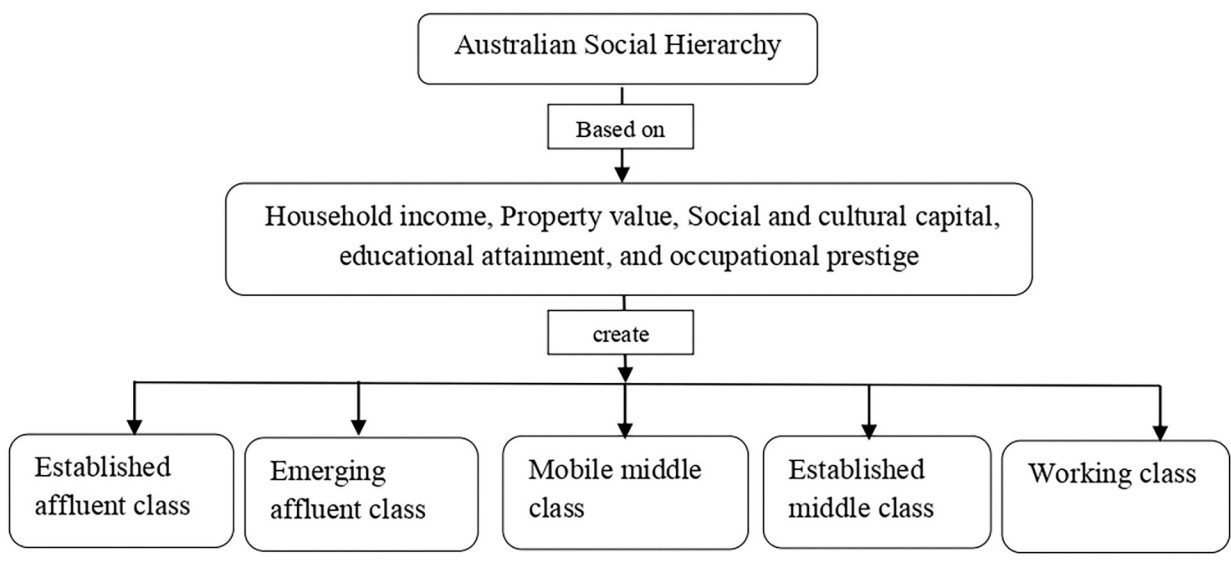

**Fig 1. Diagram of Australian social hierarchy and its' characteristics.**

Significant differences are also apparent in the average ages, education, and occupational status of the members in each class. There is evidence of intergenerational mobility in occupational prestige across all groups, particularly in the working and middle classes. This is most obvious in the two middle classes (mobile middle class and established middle class), where the prestige scores of members are more than those of their parents [27]. However, research on the application of LCA on socio-demographic variables is very scarce in Australia. To the best of the authors knowledge, the ANU report is the only study that has used this clustering approach to determine the different social classes in Australia [27]. Although one study did use data from the Australian National Health Survey (NHS) to determine the prevalence rate of mental disorders among socioeconomic groups [28], it did not use LCA to determine classes based on relevant socio-demographic factors. Notably, no study has yet shown the model-based cluster analysis of socio-demographic factors with mental health by using a nationally representative sample. To present a more comprehensive picture of the mental health of Australian children and adolescents, the current study identified different classes by considering various socio-demographic variables. These variables were taken based on the literature that shows these variables have a significant impact on the mental health of adolescents [22, 28, 29].

Therefore, the first objective of this study was to identify the cluster of the items representing socio-demographic characteristics of Australian children and adolescents aged 11–17 years by using LCA, and the second objective was to explore the link between the mental and behavioral disorders of children and adolescents with class membership and adjusted controlling factors.

## Materials and methods

### Data source and study design

Young Minds Matter The Second Australian Child and Adolescent Survey of Mental Health and Wellbeing, conducted by the University of Western Australia through the Telethon Kids Institute in collaboration with Roy Morgan Research and the Australian Government Department of Health during 2013–2014, was used to analyze cross-sectional data [30, 31]. Young Minds Matter (YMM) used a multi-stage, area-based random sampling technique that was

representative of Australian households with children and adolescents aged 4–17 years. If the household had more than one qualified child, the sample included a single child at random. In total, 6310 parents of children and adolescents aged 4–17 years (55% of eligible households) voluntarily participated in the survey through a face-to-face interview using a structured questionnaire, and 2967 adolescents aged 11–17 years (89% of eligible youth) privately completed computer-based self-reported questionnaires to provide information on their health risk behaviors (e.g., suicidality, self-harm, substance use, bullying) and service use. The survey did not include the most rural places, homeless teenagers, adolescents in any type of institutional care, or homes where interviews could not be performed in English. The survey details and methodology used in the survey can be found in Hafekost et al. [31].

## Study participants

To determine the appropriate number of classes and to examine their probable associations with adolescent mental and behavioral disorders, data were taken from parent data on the Diagnostic Interview Schedule for Children-Version IV (DISC-IV). Children and adolescents over 11 years old have a higher risk of mental disorders [32]. A study by the Telethon Kids Institute found that children and adolescents aged 12–17 years are almost three times more likely than 4–11 years old children to experience a severe mental disorder [33]. Taking these into account, in our study, the sample is restricted to children and adolescents aged 11–17 years. In parent data, there were 6,310 parents/carers involved with children aged 4–17 years. Since our study sample was restricted to adolescents 11–17 years and all the "don't know" responses were omitted, the study participants included 3,152 parents/carers.

## Measurements

**Mental disorders.**    DISC-IV was used to assess adolescent mental disorders in the 12-months preceding the YMM survey [30]. DISC-IV diagnostic criteria are referred to in the Diagnostic and Statistical Manual of Mental Disorder (DSM-IV) published by the American Psychiatric Association [34]. Worldwide, DISC-IV is considered to be the best tool for measuring the 12-month incidence of mental disorders. Primary carers completed the DISC-IV modules. Mental and behavioral disorders included major depressive disorder (MDD), attention deficit hyperactivity disorder (ADHD), conduct disorder, and four types of anxiety disorders including social phobia, separation anxiety disorder, generalized anxiety disorder, and obsessive-compulsive disorder [35].

In this study, only parental data were used to create a binary variable to detect the presence of any mental disorder, as these provided diagnostic information about each type of mental disorder in children and adolescents. The survey team created a measure of disorder severity, which has been extensively documented elsewhere in a technical report [36]. Several questions in the survey were used to indicate the presence of mental and behavioral problems or distress; while not meeting full diagnostic criteria, these may be of clinical significance or of concern to parents and carers of children and adolescents. A sub-threshold level of mental disorder was included on one or more of the DISC-IV diagnostic modules, in which symptoms were present but not at a level of severity and/or for a sufficient duration to meet diagnostic criteria. The sub-threshold level for this analysis was chosen at half or more of the number of symptoms needed to satisfy all diagnostic criteria. For example, DSM-IV criteria for MDD require the presence of six or more symptoms, and DSM-IV/5 criteria for ADHD also requires six (or more) of the associated symptoms [37, 38]. For instance, the symptoms that were considered in this study to measure MDD were as follows: "In the last year, was there a time when the child often seemed sad or depressed, seemed like nothing was fun, was grouchy or irritable,

lost weight, lost [his/her] appetite, gained a lot of weight, seemed to feel much hungrier than usual, had trouble sleeping, slept more during the day than [he/she] usually does, seemed to do things like walking or talking much more slowly than usual, seemed restless, seemed to have less energy than [he/she] usually does, feel tired, arms legs felt heavy, blamed for bad things, to do nothing well, seemed to have trouble keeping [his/her] mind on things, found it hard to make a decision, thought about death and dead people, talked seriously about killing, and making a suicide attempt". Similarly, the measurement of symptoms of other mental/behavioral disorders were considered from YMM data dictionary [39]. We then added a variable "mental and/or behavioral disorder" in our analysis to determine whether the adolescent has had any of the following types of mental and/or disorders, such as MDD, ADHD, conduct disorder, or anxiety disorders, including social phobia, separation anxiety disorder, generalized anxiety disorder, and obsessive-compulsive disorder in the past 12 months. Responses included "Yes" (if a child has at least one of these issues, coded as 1) or "No" (coded as 0).

**Socio-demographic factors.** Socio-demographic variables used in this study were represented by age (11- <15years vs. 15- <18 years), sex (male vs. female), income (<$52,000 as low, $52,000–$129,999 as medium, and >$130,000 as high), regional status (metropolitan vs. non-metropolitan), primary carer education (bachelor, diploma, and year-12/below), primary carer occupation (employed vs unemployed), index of relative socio-economic advantage and disadvantage (IRSAD) quintile [lowest (most disadvantaged), second, third, fourth and highest (most advantaged)] of the parents, family blending (intact family vs. other families) and living status of both parents in the household (yes vs. no). The IRSAD summarizes data on the economic and social circumstances of residents and households in a region, including both relative advantage and disadvantage measures. Variable "income" represents the total household income per year before tax (gross income). "Family blending" and "both parents living status in the household" indicated the family structure of the corresponding household. Families are said to be intact if both biological parents lived together and according to ABS [18], an intact family is a couple family with at least one child who is the natural or adopted child of both partners in the couple, and no step children. Non-intact families included stepfamily, blended family, single parent family and other family. In addition, the variable "both parents live in the household" indicates that children and adolescents lived with both their parents. All these socio-demographic variables were taken from three levels (i.e., individual level, household level and socio-economic level). Fig 2 represents the framework for selecting these variables to determine the significant number of classes by using LCA.

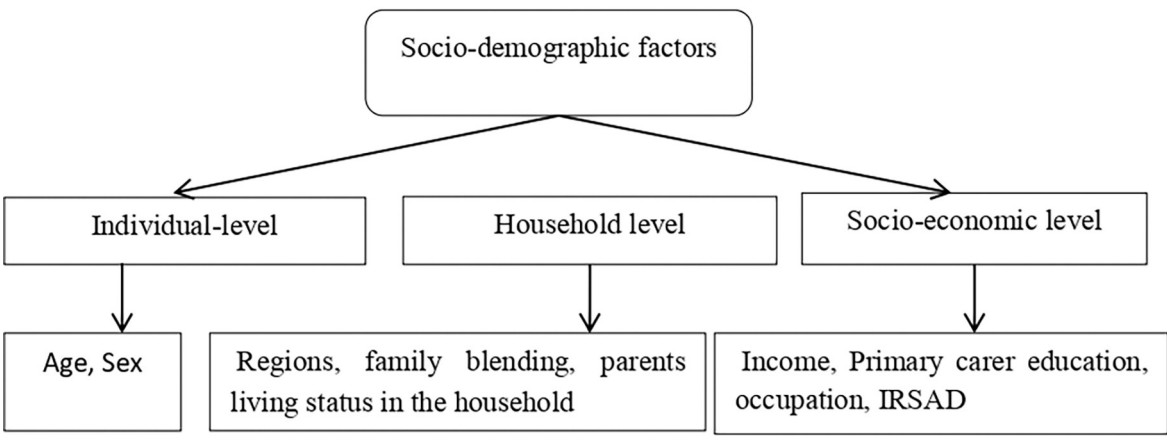

**Fig 2. Framework for latent class analysis variables.**

## Statistical analysis

**Descriptive analysis.** Descriptive analyses were carried out considering the variables that were used to find the latent classes. All variables used in this analysis were categorical. Proportions for each categorical variable were calculated using the software package R (version 4.1.3). Bivariate analyses were conducted to examine the distribution and association of the socio-demographic characteristics with the mental and behavioral disorders of children and adolescents. Chi-square tests of significance were used to describe and compare the sample characteristics of adolescents with mental and behavioral disorders. All the estimates were weighted to represent 11-17 years olds in the Australian population. In particular, the weighting took into consideration the over representation of younger children and households with more than one child between the ages of 11 and 17 as well as the oversampling component of 16- and 17-year-olds.

**Latent class analysis.** LCA is an advanced model-based clustering method that uses a probability model and is increasingly used in social, psychological, and educational research [40–42]. Like all cluster analysis methods, LCA finds homogeneous clusters of data from varied demographic and socioeconomic features by maximizing similarity within the cluster while minimizing similarity across cluster elements. LCA assumes the data are from a mixed model with several probability distributions [43]. It presumes that the data are divided into mutually exclusive homogeneous subgroups by a latent variable. LCA is said to have numerous distinct advantages over traditional cluster analysis approaches [44, 45]. Different statistical criteria are available in the LCA output, which can be used to identify the most appropriate number of clusters, and different types of variables (e.g., counts, continuous, categorical, nominal) can be used directly in LCA without any further standardization process [46].

In this study, a LCA was performed to identify and describe the classes associated with the mental and behavioral disorders of adolescents. An empirical approach was used to determine the exact number of latent classes [24]. Starting from a two-class model, the analysis was carried out consequently several times, increasing the number of subgroups by turn and replicating each of the models 5–10 times for greater precision. Then, statistical model adjustment indices such as the Bayesian information criterion (BIC), Akaike information criterion (AIC), Likelihood ratio/deviance statistic [$G^2$], and Chi-square goodness of fit were used to identify the final number of classes. The selected classes had to have enough observations to provide a representative class of a population [32]. In practice, subgroups with a size of less than 5% were not retained, as in similar studies [24]. The clusters were named in a way that best represented the most notable discoveries in the data. While naming the clusters made it easier to communicate them to the audiences [47], according to the argument, it is difficult to capture the level of difference across clusters with labels. The best potential name for each cluster's identifying traits was determined. The clusters were not meant to be displayed in a linear fashion.

**Bivariate analyses and log-binomial regression.** After the necessary number of latent classes was determined; a series of cross-tabulations and bivariate analyses (using Chi-square tests) were performed to investigate the distribution of and relationships between the mental and behavioral disorders of children and adolescents. The generalized linear model with the log-link binomial family (log-binomial regression model) was taken into consideration in this study due to the high prevalence of mental and behavioral disorders [48]. Two models were fitted to observe the association between class membership and the mental health status of children and adolescents and to estimate the prevalence ratio. The first model was considered without controlling any covariate and thus provided an unadjusted effect. The second model was based on considering the controlling variable of "household size" and thus illustrated adjusted effects.

### Ethics statement

This study used secondary data from the YMM survey dataset. All participants provided written consent to participate in the survey, which was voluntary. Ethical approval was obtained from the Human Research Ethics Committee of the University of Southern Queensland to conduct this research. University gave us the written approval through email based on our Human Research Ethics (HRE) application.

## Results

The socio-demographic characteristics of the study participants are presented in Table 1. According to age group, 51.1% of the total samples were between 15 and <18 years old and the mean age was 14.6 years. The male/female distribution was equitable, with 51.9% males and 48.1% females. Regional status displayed that 61.8% of the children and adolescents lived in the metropolitan area and 61.3% came from intact families. The study results also showed that 66.1% of the children and adolescents lived with both of their parents. In the socio-economic level variables, about 44.1% of households belonged to the medium income group ($52000–$129999), 28.9% fell in the high-income group, and 27.0% in the low-income group. In parental education from the socio-economic level, 30.4% of parents had the highest level of education (bachelor), around 37% had a diploma and around 32% had the lowest level of education (year 12/below). Most parents of children and adolescents were employed (75.1%) and 24.9% were unemployed. Sample data also showed that 22.5% of households belong to the highest level in the IRSAD quintile and 16.1% was in the lowest level. A total of 38.5% of children and adolescents in this study population had mental and behavioral disorders.

The study results also revealed that all the socio-demographic variables considered in this research were significantly associated with the mental and behavioral disorders of adolescents (see S1 Table).

### Identification of latent classes and associated characteristics

**Model selection by latent class analysis.** The findings from the LCA are presented in Table 2. The BIC and G^2 values suggested a minimum for a five-class distribution. Thus, the five-class model was selected as the best fitting model based on the entire set of data, even though the AIC and Pearson $\chi^2$ values decreased as the number of classes increased.

### Characteristics of the identified latent classes

Table 3 represents the characteristics of each class. Among the five classes, adolescents who belong to the most privileged group were classified as class 5 (highest socio-economic status and intact family structure), comprising 23.3% (734) of the total sample. In this class, more than 72% of households belong to the high-income group (against 3.8% in class1, 6.2% in class 2, 28.8% in class 3, and 24.0% in class 4), and around 87% of them live in the metropolitan area. More than 59% of the parents in class 5 have the highest level of education, and about 85% of the parents are employed. Most of the children and adolescents (around 98%) in this cluster lived with both of their parents and about 95% of them came from intact families. Another important characteristic of this class is that about 64.6% of households belong to the most advantaged level of the variable IRSAD quintile and there is no one in this group belonging to the lowest two disadvantaged levels.

On the other hand, adolescents who belong to the underprivileged group were classified as class 1 (lowest socio-economic status and non-intact family structure), representing 19.7% (n = 621) of the total sample. In this class, about 55% of children and adolescents lived in non-

**Table 1. Socio-demographic characteristics of the study participants.**

| Characteristics | | | Frequency (n) (unweighted) | % (weighted) |
|---|---|---|---|---|
| Individual-level | Age (mean ± SD:14.6 ± 2.04) | 11 to <15 years | 1,393 | 48.9 |
| | | 15 to <18 years | 1,759 | 51.1 |
| | Sex | Male | 1,636 | 51.9 |
| | | Female | 1,516 | 48.1 |
| Household-level | Regional status [a] | Metropolitan | 1,977 | 61.8 |
| | | Non-metropolitan | 1,175 | 38.2 |
| | Family blending [b] | Intact family | 1,996 | 61.3 |
| | | Other families | 1,156 | 38.7 |
| | Both parents live in the household | Yes | 2,147 | 66.1 |
| | | No | 1,005 | 33.9 |
| Socio-economic level | Household income/year | Low | 760 | 27.0 |
| | | Middle | 1,460 | 44.1 |
| | | High | 932 | 28.9 |
| | Parental education | Bachelor | 976 | 30.4 |
| | | Diploma | 1,165 | 37.5 |
| | | Year 12/below | 1,011 | 32.1 |
| | Parental employment | Employed | 2,370 | 75.1 |
| | | Unemployed | 782 | 24.9 |
| | IRSAD quintile [c] | Lowest | 493 | 16.1 |
| | | Second | 557 | 18.0 |
| | | Third | 666 | 21.8 |
| | | Fourth | 691 | 21.7 |
| | | Highest | 745 | 22.5 |

n = unweighted number of respondents; % = weighted percentage.

[a] Regional status: Regional status indicates whether adolescents come from the metropolitan area or non-metropolitan area.

[b] Family blending: Intact families refer to those adolescents who come from families in which their parents are married and live together. Other families include step, blended, and single-parent families.

[c] IRSAD: "The Index of Relative Socio-economic Advantage and Disadvantage (IRSAD): "It presents statistics on people's and households' economic and social circumstances in a given area, including relative advantage and disadvantage indicators. A low score suggests a relative lack of advantage, whereas a high score shows a relative lack of disadvantage and greater advantage.

metropolitan areas and around 76% of the household belongs to the low-income group. More than 51% of parents in this class were unemployed, and approximately the same percentage (49.2%) of parental educational status belongs to the lowest level (year12/below). The most

**Table 2. Information for determining the latent class model and the fit statistic.**

| No. of Classes | AIC | BIC | G^2 | $\chi^2$ | Estimated class population shares | | | | | |
|---|---|---|---|---|---|---|---|---|---|---|
| | | | | | class 1 | class 2 | class 3 | class 4 | class 5 | class 6 |
| 1 | 48168.9 | 48253.7 | 7120.1 | 18129.0 | 100% | | | | | |
| 2 | 44697.0 | 44872.7 | 3618.3 | 5519.9 | 33.8% | 66.2% | | | | |
| 3 | 44093.2 | 44359.6 | 2984.4 | 4303.5 | 27.7% | 39.3% | 33.0% | | | |
| 4 | 43619.6 | 43976.9 | 2480.8 | 3266.0 | 13.3% | 26.9% | 19.9% | 39.9% | | |
| 5 | 43446.4 | 43894.5 | 2277.6 | 2887.9 | 19.6% | 16.7% | 27.6% | 12.7% | 23.3% | |
| 6 | 43379.6 | 43918.6 | 2280.8 | 2797.9 | 16.8% | 11.5% | 27.7% | 9.0% | 23.3% | 11.8% |

Note: AIC-Akaike Information Criterion; BIC-Bayesian Information Criterion; G^2 [Likelihood ratio/deviance statistic]; $\chi^2$ [Chi-square goodness of fit].

**Table 3. Characteristics of class obtained in adolescents aged 11–17 in Australia.**

| | | Lowest socio-economic status and non-intact family structure (Class 1) (%) | Worst education and occupational attainment and intact family structure (Class 2) (%) | Average socio-economic status and intact family structure (Class 3) (%) | Good socio-economic status and non-intact family structure (Class 4) (%) | Highest socio-economic status and intact family structure (Class 5) (%) |
|---|---|---|---|---|---|---|
| | Total proportion | 19.75 | 16.7 | 27.60 | 12.70 | 23.30 |
| | n (total = 3152) | 621 | 526 | 870 | 400 | 734 |
| **Socio-demographic characteristics** | | | | | | |
| **Individual-level** | **Age** | | | | | |
| | 11 to <15 years | 46.18 | 46.33 | 44.88 | 37.67 | 43.72 |
| | 15 to <18 years | 53.82 | 53.67 | 55.12 | 62.33 | 56.28 |
| | **Sex** | | | | | |
| | Male | 53.74 | 50.60 | 52.54 | 48.18 | 52.55 |
| | Female | 46.26 | 49.40 | 47.46 | 51.82 | 47.45 |
| **Household-level** | **Regional status** | | | | | |
| | Metropolitan | 44.99 | 63.83 | 49.89 | 72.14 | 87.08 |
| | Non-metropolitan | 55.01 | 36.17 | 50.11 | 27.86 | 12.92 |
| | **Family blending** | | | | | |
| | Intact family | 1.72 | 94.05 | 91.22 | 000 | 94.86 |
| | Other families | 98.28 | 5.95 | 8.78 | 100 | 5.14 |
| | **Both parents living in the household** | | | | | |
| | Yes | 6.24 | 98.91 | 99.5 | 000 | 98.24 |
| | No | 93.76 | 1.09 | 0.5 | 100 | 1.76 |
| **Socio-economic level** | **Household income/year** | | | | | |
| | Low | 75.65 | 34.85 | 5.56 | 11.68 | 1.44 |
| | Medium | 20.54 | 59.97 | 65.56 | 64.29 | 26.54 |
| | High | 3.82 | 6.19 | 28.88 | 24.02 | 72.02 |
| | **Parental education** | | | | | |
| | Bachelor | 5.80 | 21.20 | 24.89 | 44.34 | 59.25 |
| | Diploma | 44.96 | 23.09 | 48.79 | 40.41 | 24.22 |
| | Year 12/below | 49.23 | 55.71 | 26.32 | 15.24 | 16.53 |
| | **Parental employment** | | | | | |
| | Employed | 48.49 | 37.80 | 100 | 94.03 | 85.00 |
| | Unemployed | 51.51 | 62.20 | 000 | 5.97 | 15.00 |
| | **IRSAD** | | | | | |
| | Lowest (most disadvantages) | 38.67 | 23.72 | 12.00 | 5.74 | 000 |
| | Second | 25.47 | 23.70 | 25.40 | 13.18 | 000 |
| | Third | 20.73 | 28.10 | 24.41 | 22.71 | 11.73 |
| | Fourth | 10.46 | 17.22 | 29.57 | 26.02 | 23.72 |
| | Highest (most advantages) | 4.68 | 7.25 | 8.61 | 32.36 | 64.55 |

important characteristic of this cluster is that only 4.6% of the household belongs to the most advantaged group and around 38.7% belongs to the most disadvantaged group in the IRSAD quintile. In this class, only 6.2% of children and adolescents lived with both of their parents and fewer than 2% came from intact families.

Class 2 is defined by those adolescents who belong to a group with the worst education and occupational attainment and intact family structure, which is better than the underprivileged

group of class 1. This class represented 16.7% (n = 526) of the total sample, where only 6.2% of household belongs to the high-income group and about 24% belongs to the most disadvantaged level in the IRSAD quintile. In addition, this class had the highest percentage of parents with the lowest level of education (55.7%) and most of the parents were unemployed (62.2%).

Class 3 (average socio-economic status and intact family structure) and Class 4 (good socio-economic status and non-intact family structure) were characterized by those individuals who belong to moderate privileged and privileged groups, consisting respectively of 27.6% (n = 870) and 12.7% (400) of the total sample. More than 72% of the households of class 4 lived in the metropolitan area and this proportion was reduced to 50% in the case of class 3.

Although the proportion of high and middle-income levels was higher for class 3 compared to class 4, in the IRSAD quintile about 32.4% of the household in class 4 belongs to the most advantaged level, while in class 3, only 8.6% households belong to this level. Most of the households (79.4%) in this class 3 also fall between the most disadvantaged and advantaged levels in the IRSAD quintile. More than 44% of the parents of class 4 have the highest level of education, and about 94% of the parents in this class were employed. In class 3, around 49% of parents have a medium level of education (diploma) and 100% were employed. Moreover, 65.56% of the households' annual income in class 3 belongs to the medium-income level. One of the important disadvantages of class 4 was that no children and adolescents in this class came from intact families and at least one or no children and adolescents lived with both of their parents. All these five classes have some common characteristics. For instance, more than 50% of children and adolescents in these classes (except class 4) were male and a greater proportion of children and adolescents in all five classes were from the older age group.

## Prevalence of mental and behavioral disorders by classes

Table 4 shows that the prevalence of mental and behavioral disorders of class 1 was 52.4%. This means that, if all the respondents were from class 1, then the average prevalence rate of mental and/or behavioral disorders would have been 0.52. Class 4 shared the lowest class membership (12.7%, see in Table 3) but it had a greater prevalence (44.1%) of mental and/or behavioral disorders than did class 2 (35.2%), class 3 (32.9%), and class 5 (32.7%). As a result, it can be deduced that if all respondents had been from class 4, then the prevalence rate of mental and/or behavioral disorders would have been 0.44 on average. In addition, the second highest percentage of adolescents (23.3%, see in Table 3) belongs to class 5, only 32.7% of whom suffered from mental and behavioral disorders.

**Table 4. Distribution of mental and behavioral disorder by class membership.**

| Latent classes | Mental and/or behavioral disorders |
|---|---|
| | Yes n (%) |
| Class 1 (Lowest socio-economic status and non-intact family structure) | 330 (52.4) |
| Class 2 (Worst education and occupational attainment and intact family structure) | 148 (35.2) |
| Class 3 (Average socio-economic status and intact family structure) | 320 (32.9) |
| Class 4 (Good socio-economic status and non-intact family structure) | 174 (44.1) |
| Class 5 (Highest socio-economic status and intact family structure) | 240 (32.7) |
| Total | 1,212 (38.5) |

p-value: <0.001.

**Table 5. Log binomial regression model of predicting latent class membership (unadjusted and adjusted) and control variables.**

| Latent classes (ref. Highest socio-economic status and intact family structure (Class 5)) | Unadjusted | | Adjusted[a] | |
|---|---|---|---|---|
| | Coefficient | PR (95% CI of PR) | Coefficient | PR (95% CI of PR) |
| Good socio-economic status and non-intact t family structure (Class 4) | 0.30*** | 1.35 (1.16, 1.57) | 0.29*** | 1.35 (1.15, 1.56) |
| Average socio-economic status and intact family structure (Class 3) | 0.01 | 1.01 (0.87, 1.15) | 0.01 | 1.01 (0.88, 1.15) |
| Worst education and occupational attainment and intact family structure (Class 2) | 0.07 | 1.08 (0.91, 1.27) | 0.08 | 1.09 (0.92, 1.28) |
| Lowest socio-economic status and non-intact family structure (Class 1) | 0.47*** | 1.60 (1.41, 1.82) | 0.47*** | 1.60 (1.41, 1.82) |

Note: PR- Prevalence ratio; CI-Confidence interval; *: $p<0.05$; **: $p<0.01$; ***: $p<0.001$.

[a] Control variable: "Household size".

## Regression modeling

Table 5 presents the results of log-binomial regressions examining the associations between identified classes and mental and behavioral disorders after adjusting for covariates. Children and adolescents belonging to class 1 (lowest socio-economic status and non-intact family structure) and class 4 (good socio-economic status and non-intact family structure) were about 1.60 and 1.35 times more prevalent to be suffering from mental and/or behavioral disorders compared to their class 5 (highest socio-economic status and good family structure) counterparts (95% CI of PR: 1.41–1.82 for class 1, and 95% CI of PR: 1.16–1.57 for class 4). Class 2 (worst education and occupational attainment and intact family structure) and class 3 (average socio-economic status and intact family structure) also showed higher prevalence of mental and behavioral disorders than did class 5, but they were not statistically significant.

An analysis introducing a control variable (household size) demonstrated similar results as the unadjusted model. Specifically, classes 1 and 4 in this adjusted model showed significantly higher prevalence ratio of mental and behavioral disorders than did class 5. Thus, it can be stated that class 1 (lowest socio-economic status and non-intact family structure) and class 4 (good socio-economic status and non-intact family structure) were the most vulnerable classes in the context of suffering from mental and behavioral disorders.

## Discussion

In this study, LCA was used to identify distinct classes based on various socio-demographic factors of Australian children and adolescents. Patterns of socio-economic and demographic features were characterized by various levels of engagement in social behaviors across multiple domains and were differentially associated with mental and behavioral disorders [10, 19, 22, 28]. Differences in socio-demographic characteristics and mental health issues were observed in the heterogeneous classes, supporting the view that structural factors (e.g., age, gender, region, occupation, education, and family blending) influence the mental health of individuals. Using nine distinct socio-demographic factors, we were able to identify 5 latent classes in our study. Mental and behavioral disorders of children and adolescents were mostly prevalent in class 1 and rarely in class 5. Although, the lowest percentage of children and adolescents was from class 4, the prevalence of mental and behavioral disorders in this class was higher compared to that in classes 2, 3, and 5. The model of log-binomial regression revealed that, in comparison to class 5, classes 1 and 4 showed higher odds of mental and behavioral disorders even after adjusting for control variables.

Class 1 in this study represents the characteristics of the lowest level of income, education, and occupation. Most of the households in this class 1 belong to the most disadvantaged level in the IRSAD quintile. This result is consistent with the findings of ANUpoll report [20],

which showed that, out of five social classes, class 1 (established working class) has the lowest incomes, occupational prestige, and educational attainment. Class 2 has the worst level of parental education and employment status compared to the other four classes, and the second-highest percentage of adolescents of this class belong to the most disadvantaged level in the IRSAD quintile. Australians in class 3, consisting of 25% of the population, have higher levels of educational qualifications and consequently have higher household income [27]. However, in our study, the highest percentage of parental educational attainment and most of the households' annual income belongs to the medium level in class 3. Class 4 shared many of the characteristics of class 3, with three key differences including family blending, living status of both parents in the household, and the IRSAD quintile. The main disadvantage of this class was that no children and adolescents had come from intact families, and no one lived with both parents, thereby indicating relatively poor family functioning. Class 5 possesses the highest level of income, education, and intact family structures. Most of the households of this class belong to the most advantaged level in the 'IRSAD-quintile'. In our study, we found that most children and adolescents in classes 1 and 4 lived in non-intact families which is more often associated with poorer family functioning. According to past research, good family functioning has a positive impact on the mental health of children and adolescents [19, 49]. Therefore, instead of only focusing on individual socio-demographic characteristics [14, 21, 33, 41], our findings suggested underlying clusters of children and adolescents at risk for mental and behavioral disorders, specifically, those who were from an underprivileged group. Previous studies showed that children and adolescents from the socio-economically disadvantaged group were two to three times more likely to suffer from mental health issues than did their counterparts who were from the socio-economically advantaged groups [15]. Daniel et al. (2015) revealed that adolescents in intact families evaluated better family functioning and parental–child relationship qualities than adolescents in non-intact families [33]. In 2015, another study found that adolescents from non-intact households had a lower perception of family functioning, behavioral control of both parents, paternal psychological control, and parent-child relational traits than did adolescents from intact households [19]. The ABS report on Australian families with children and adolescents (2011) showed that the lowest percentage of adolescents 15–17 years old lived with both of their natural parents [13]. Interestingly, in our study, a greater proportion of adolescents who were suffering from mental and behavioral disorders belong to this aged group (15–17 years). Due to the non-intact family structures, even though members of class 4 were socio-economically advantaged, mental and behavioral disorders among children and adolescents in this class were more prevalent compared to those in classes 2 and 3. The present study expands the existing literature by providing evidence that apart from individual poor socio-economic conditions, the family structure with poor family functioning can be associated with the mental and behavioral disorders of children and adolescents. Thus, to enhance mental health, particularly among children and adolescents living in non-intact families and in households with low socioeconomic status, health promotion and prevention as well as tackling poverty are necessary.

## Strengths and limitations

The strengths of this study included the use of person-centered approach (i.e., LCA) that concurrently examines the socio-demographic characteristics of children and adolescents and determines the associations between obtained latent classes and their mental and behavioral disorders by using nationally representative YMM data. This study has a few limitations that need to be mentioned. First, the 55% response rate of the survey could have created some bias that weighting did not take into consideration. Second, the prevalence of mental and/or

behavioral disorders is comparatively higher in our study because we considered the partial criteria for confirming the existence of specific mental and/or behavioral problems (such as ADHD) by taking into consideration all of their symptoms. The main reason for doing that was to determine the class-wise prevalence of mental and/or behavioral disorders in children and adolescents based on their initial symptoms to facilitate early diagnosis. According to our study objectives, we intended to investigate the relationship between mental and behavioral disorders of children and adolescents and various latent classes in order to distinguish vulnerable classes from privileged classes. As is widely known, some mental and/or behavioral disorders are the most commonly diagnosed disorders in children and adolescents, therefore, it is important to ascertain their prevalence by class at the earliest possible stage to facilitate early diagnosis and treatment." However, over-diagnosing mental issues might lead to mis-medication and social stigmas. Third, this study explored the association of the mental and behavioral disorders of children and adolescents with latent classes, but we did not show the relationship between latent classes with various kinds of mental and behavioral disorders in children and adolescents. Fourth, LCA allocated individuals to classes based on their likelihood of belonging to a class given their patterns of indicator scores. However, there is no certainty that the class assignments were completed accurately. Moreover, we did not consider the mental health status of the parents/carers may have an impact on the outcome of our study. "Don't know" responses were omitted in our study, which may have a significant impact on the result. Lastly, the analysis was based on a cross-sectional study and self-reported responses, where causal relationships cannot be established through a cross-sectional study, and self-reported responses compromise the reliability and validity of measurement.

## Conclusions

The present study has some important public health implications as it has identified the cluster of items that represent socio-demographic characteristics and revealed their associations with the mental and behavioral disorders of children and adolescents. The significant impact of individual class on the mental and behavioral disorders will help government and non-government agencies as well as practitioners differentiate the vulnerable classes from privileged classes in formulating health-related policy and strategy. The findings indicate that health promotion and prevention as well as combating poverty are needed to improve mental health among children and adolescents living in non-intact families and in families with a low socio-economic status.

## Supporting information

**S1 Table. Association of socio-demographic characteristics with mental and/or behavioral disorders of children and adolescents.**
(DOCX)

## Acknowledgments

The authors would like to thank the Telethon Kids Institute, The University of Western Australia, Roy Morgan Research, the Australian Government Department of Health for conducting the survey, and the Australian Data Archive for providing the YMM data set. The authors would also like to express their gratitude to Dr Barbara Harmes for proofreading the manuscript before submission.

## Author Contributions

**Conceptualization:** Nahida Afroz.

**Data curation:** Nahida Afroz.

**Formal analysis:** Nahida Afroz.

**Methodology:** Nahida Afroz, Enamul Kabir.

**Supervision:** Enamul Kabir, Khorshed Alam.

**Writing – original draft:** Nahida Afroz.

**Writing – review & editing:** Enamul Kabir, Khorshed Alam.

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
