## [Decision Letter · Decision Letter 0]

27 Jan 2023

PONE-D-22-28764Clustering of socio-demographic factors and their association with the mental health of Australian children and adolescents: A latent class analysisPLOS ONE

Dear Dr. Afroz,

Thank you for submitting your manuscript to PLOS ONE. After careful consideration, we feel that it has merit but does not fully meet PLOS ONE’s publication criteria as it currently stands. Therefore, we invite you to submit a revised version of the manuscript that addresses the points raised during the review process. The revised version should address all comments.

 Please submit your revised manuscript by Mar 13 2023 11:59PM. If you will need more time than this to complete your revisions, please reply to this message or contact the journal office at plosone@plos.org. Please include the following items when submitting your revised manuscript:A rebuttal letter that responds to each point raised by the academic editor and reviewer(s). You should upload this letter as a separate file labeled 'Response to Reviewers'.A marked-up copy of your manuscript that highlights changes made to the original version. You should upload this as a separate file labeled 'Revised Manuscript with Track Changes'.An unmarked version of your revised paper without tracked changes. You should upload this as a separate file labeled 'Manuscript'.

We look forward to receiving your revised manuscript.

Kind regards,

Petri Böckerman

Academic Editor

PLOS ONE

Journal Requirements:

3. "Please update your submission to use the PLOS LaTeX template. The template and more information on our requirements for LaTeX submissions can be found at " ext-link-type="uri" xlink:type="simple">http://journals.plos.org/plosone/s/latex."

Additional Editor Comments (if provided):

The revised version should address all comments.

Reviewers' comments:

Reviewer's Responses to Questions

**Comments to the Author**

1. Is the manuscript technically sound, and do the data support the conclusions?

Reviewer #1: Yes

Reviewer #2: Partly

2. Has the statistical analysis been performed appropriately and rigorously? 

Reviewer #1: Yes

Reviewer #2: No

3. Have the authors made all data underlying the findings in their manuscript fully available?

Reviewer #1: Yes

Reviewer #2: No

4. Is the manuscript presented in an intelligible fashion and written in standard English?

Reviewer #1: Yes

Reviewer #2: No

5. Review Comments to the Author

Reviewer #1: Major points

1. Study rationale. Why would it be important to use LCA to study SES? In the abstract it reads that ” However, no study has yet been conducted on a model-based cluster analysis of socio-demographic characteristics with mental health.”, but why would be important to use certain statistical technique?

2. Literature review is rather short. There is huge amount of literature examing the association between childhood socioeconomic environment and later risk of mental disorders using large scale registry and survey data. See for example:

https://doi.org/10.1093/ije/dyab066

https://doi.org/10.1186/s12916-020-01794-5

https://doi.org/10.1016/j.socscimed.2016.12.040

3. Attrition analysis is missing.

4. Study limitations are completely missing. LCA can produce unstable results, which are difficult to replicate in other datasets. Study design was cross-sectional. Moreover, can reverse causality, child influence parental SES also be possible? Anyhow, presents results don’t have causal interpretation.

5. Very different kind of mental disorders very grouped together. Please conduct sensitivity analyses were different type of mental disorders are analysed separately. In addition, please report prevalence of different type of mental disorders in girls and boys.

6. Discussion is difficult to follow as results are repeated in detail. Please first state your primary results and then discuss them in the context of previous studies.

7. Potential mechanisms explaining the study findings are currently missing.

Minor points

Two decimals should be enough in text and in tables

p-value can’t be 0.000 (table 2)

Reviewer #2: Overall, I find the manuscript objective and the methodological approach very interesting. From my point of view, however, the manuscript needs a major revision. In particular, the operationalization of "mental health" should be in accordance with the standards. The authors reported a 12month prevalence of 63.5 % for mental disorders in adolescents. In the manuscript of Lawrence et al. (doi:10.1177/0004867415617836), who used the same data, the 12month prevalence of mental disorders was 13.9 %.

6. PLOS authors have the option to publish the peer review history of their article (what does this mean?). If published, this will include your full peer review and any attached files.

Reviewer #1: No

Reviewer #2: **Yes: **Petra Rattay

---

## [Author Response · Author response to Decision Letter 0]

14 Mar 2023

Editor’s comments:

After careful consideration, we feel that it has merit but does not fully meet PLOS ONE’s publication criteria as it currently stands. Therefore, we invite you to submit a revised version of the manuscript that addresses the points raised during the review process.

• A rebuttal letter that responds to each point raised by the academic editor and reviewer(s). You should upload this letter as a separate file labelled 'Response to Reviewers'.

• A marked-up copy of your manuscript that highlights changes made to the original version. You should upload this as a separate file labelled 'Revised Manuscript with Track Changes'.

An unmarked version of your revised paper without tracked changes. You should upload this as a separate file labelled 'Manuscript'.

Author’s response: Thank you so much for giving us the opportunity to submit a revised version of the manuscript. We have addressed all the comments and hope the quality of this paper has now been improved substantially. Please find attached our revised manuscript (a marked-up copy labelled ‘Revised Manuscript with Track Changes’ and an unmarked version labelled 'Manuscript') and below our point-by-point response to all reviewers.

Reviewers# 1 comments

Major comments

1.Study rationale. Why would it be important to use LCA to study SES?

 In the abstract it reads that ” However, no study has yet been conducted on a model-based cluster analysis of socio-demographic characteristics with mental health.”, but why would be important to use certain statistical technique?

Author’s response:Thanks for asking these questions.

In this research, we did not use LCA to study socioeconomic status only. Rather we used it to identify the sociodemographic classes based on three different levels, namely individual level, household level, and socio-economic level. This classification is based on the landmark report of the ANUpoll 2015 which showed that Australian population can be divided into five distinct classes depending on their income and wealth, regional status, employment, occupation, family bonding, and other factors [21].

The authors of this study considered model-based cluster analysis for the following reasons:

Latent Class Analysis (LCA) is an advanced model-based clustering method that uses a probability model and is increasingly being used in social, psychological, and educational research [35–37]. LCA offers a powerful analytical approach for categorizing groups (or ‘‘classes’’) within a heterogenous population and identifies these hidden classes by a set of predefined features. Unlike many other grouping analytical approaches, LCA derives classes using a probabilistic approach[38]. Therefore, instead of analyzing each socio-demographic factor separately, in this study, we have considered a latent class analysis (LCA), which allows for a person-centered approach to understanding how a wide range of socio-demographic factors may cluster together to reflect distinct social classes within a large and heterogeneous population.

2. Literature review is rather short. There is huge amount of literature examing the association between childhood socioeconomic environment and later risk of mental disorders using large scale registry and survey data. 

Author’s response: Thank you for notifying this issue and suggesting the relevant articles.

We have revised the literature review section and included more background information to justify the study as follows:

“The family structure is a representation of the connection between one's living arrangements, marital status, and biological relatedness[13].According to the Australian Bureau of Statistics (ABS) report of Australian families with children and adolescents, 89% of families were “intact”, 6% were stepfamilies and 5% were blended families. In addition, 71% of all children lived with both natural parents (bio-logical parents), 4% were in stepfamilies, and 5% were in blended families. This report also stated that the percentage of adolescents living in their household varies with their age. Naturally, children under the age of five were more likely to live with both of their biological parents (78%), followed by those between the ages of 5-9 years (72%). The least likely to be in this scenario (63%) were those between the ages of 15 and 17 years [14].”

“Daundasekara et. al., (2022) used LCA to identify socio-economic and health risk profiles among mothers of young children predicting longitudinal risk of food insecurity [20]”.

In addition, according to the suggestions of reviewers 2, the description of the social hierarchy of Australia and the “Diagram of Australian social hierarchy and its characteristics” have been presented in the background section instead of methods section.

3. Attrition analysis is missing.

Author’s response: Thank you for raising this issue.

We know that attrition is one of the key methodological concerns in longitudinal or cohort studies. Cross sectional studies evaluate variables at single point in time, therefore, respondents are less likely to drop out once agreed to participate. In cross-sectional survey design, response rate or completion rate is considered to be important for ensuring data quality and statistical precision. The survey data used in the current study achieved a response rate of 55%. Also, the number of cases in this study is 3152, which is much greater than in a nationally representative sample. As a result, we did not examine attrition analysis, although we have noted this in the limitation.

4. Study limitations are completely missing. LCA can produce unstable results, which are difficult to replicate in other datasets. Study design was cross-sectional. Moreover, can reverse causality, child influence parental SES also be possible? Anyhow, presents results don’t have causal interpretation.

Author’s response: Thank you so much for pointing out this issue. 

With the strength of the current research, this study has a few limitations. We have included this in our revised manuscript as follows: 

“Strengths and limitations

The strengths of this study included the use of person-centered approach (i.e., LCA) that concurrently examines the socio-demographic characteristics of children and adolescents and determines the associations between obtained latent classes and their mental and behavioral disorders by using nationally representative YMM data. This study has a few limitations that need to be mentioned. First, the 55% response rate of the survey could have created some bias that weighting did not take into consideration. Second, this study explored the association of the mental and behavioral disorders of children and adolescents with latent classes, but we did not show the relationship between latent classes with various kinds of mental and behavioral disorders in children and adolescents. Moreover, we did not consider the mental health status of the parents/carers may have an impact on the outcome of our study. “Don’t know” responses were omitted in our study, which may have a significant impact on the result. Lastly, the analysis was based on a cross-sectional study and self-reported responses, where causal relationships cannot be established through a cross-sectional study, and

5. Very different kind of mental disorders very grouped together. Please conduct sensitivity analyses were different type of mental disorders are analysed separately. In addition, please report prevalence of different type of mental disorders in girls and boys.

Author’s response: Thank you for raising this issue.

“The objectives of our study were to identify the cluster of the items representing socio-demographic characteristics of Australian children and adolescents and then explored the link between the mental and/or behavioral disorders of children and adolescents with class membership and adjusted controlling factors. 

Therefore, we have only considered the variable “mental and/or behavioural disorder” in our analysis (whether children and adolescents had any major types of mental and/or behavioral disorders or not), and we did not consider different types of mental disorders separately. This is one of our future objectives and have a plan to write another paper based on this and other objectives.

In addition, according to Shanahan et al. (2013), any particular latent class should not contain less than 5% of the sample [18]. In that case, performing LCA for different types of mental disorders individually could lead to results that are deceptive and/or fall short of LCA requirements. 

6. Discussion is difficult to follow as results are repeated in detail. Please first state your primary results and then discuss them in the context of previous studies.

Author’s response: Thank you for your valuable recommendations. The authors have revised the discussion section according to the reviewer’s suggestion. We hope that the clarity of the section has been improved.

7. Potential mechanisms explaining the study findings are currently missing.

Author’s response: Thanks for pointing out this issue.

We have explained how the study findings may be important for policy and practice in the revised version of the manuscript.

Minor points: Two decimals should be enough in text and in tables. p-value can’t be 0.000 (table 2)

Author’s response: Thank you for your feedback.

We have considered two decimal points in all our tables 1-6.

 P-value has been written as 0.001 instead of 0.000 in the revised manuscript.

Reviewers# 2 comments

1. Overall, I find the manuscript objective and the methodological approach very interesting. From my point of view, however, the manuscript needs a major revision. In particular, the operationalization of "mental health" should be in accordance with the standards. The authors reported a 12month prevalence of 63.5 % for mental disorders in adolescents. In the manuscript of Lawrence et al. (doi:10.1177/0004867415617836), who used the same data, the 12month prevalence of mental disorders was 13.9 %.

Author’s response: Thanks to the reviewer for providing valuable feedback on the submitted paper. 

And thanks again for pointing out this major issues. In a previous study, Irteja et al. (https://doi.org/10.1371/journal.pone.0231180) using this YMM data showed that the 12-month prevalence rate of mental disorders of adolescents aged 13-17 years was 34.7%. However, the authors did not clearly explain how they measured mental disorders. In our study, as we include children as young as 11, we investigated at the age range of 11-17 years in our study. While we measured each mental disorder, we created a binary variable and considered the presence of the corresponding mental disorder when at least one of the relevant symptoms was present. Consequently, we then added a variable ‘mental disorder’ in our analysis: whether the adolescent has had any of the following types of mental disorders—anxiety disorders, major depressive disorder, ADHD and conduct disorder. As a result, the prevalence rate was a bit high in our study. However, the distribution of the proportion of mental disorders by various sociodemographic factors in our study showed consistent results with the previous study. For instance, the prevalence of mental disorders was higher in the age group 15 to 17, which was consistent with the findings of Irteja et al. (2020) [43]. Similarly, the prevalence of mental disorders with other socio-demographic factors such as regional status, family blending/ family type, household income, parental education, occupation, employment status, and IRSAD quintile also showed quite similar results with the prior studies. 

After considering your recommendation we have deeply reviewed some past studies including Lawrence et al. (2016) to better understand the criteria of the presence of a specific disorder. After that, we have framed the variable “Mental and/or behavioural disorder” as follows:

 Several questions in the survey were used to indicate the presence of mental or behavioural problems or distress while not meeting full diagnostic criteria, may be of clinical significance or of concern to parents and carers or children and adolescents. A sub‐threshold level of mental disorder on one or more of the DISC‐IV diagnostic modules, in which symptoms are present but not at a level of severity and/or for a sufficient duration to meet diagnostic criteria. For this analysis, the sub‐threshold level was set at half or more of the required number of symptoms to meet full diagnostic criteria. For example, DSM‐IV criteria for major depressive disorder require the presence of 6 or more symptoms, and DSM-IV/5 criteria for ADHD also requires six (or more) of the associated symptoms. The symptoms that were considered in this study for measuring MDD, for example, were- In the last year, was there a time when CHILD often seemed sad or depressed?, seemed like nothing was fun?, was grouchy or irritable?, lost weight?, lose [his/her] appetite?, gained a lot of weight?, seemed to feel much hungrier than usual?, had trouble sleeping?, slept more during the day than [he/she] usually does?, seemed to do things like walking or talking much more slowly than usual, seemed restless?, seemed to have less energy than [he/she] usually does?, feel tired?, arms legs felt heavy?, blamed for bad things?, nothing do well?, seemed to have trouble keeping [his/her] mind on things?, hard to make decision?, thought about death and died people?, talk seriously about killing?, and take suicide attempt? Similarly, for measuring other mental/behavioural disorders, we have considered the relevant questions about the symptoms of a particular disorders from YMM data dictionary. 

We then added a variable ‘mental and/or behavioural disorder’ in our analysis: whether the adolescent has had any of the following types of mental and/or behavioural disorders - major depressive disorder, ADHD, conduct disorder, or anxiety disorders which includes social phobia, separation anxiety disorder, generalized anxiety disorder, and obsessive-compulsive disorder in the past 12 months. Responses included ‘Yes’ (if a child has at least one of these issues, the code is 1) or ‘No’ (coded as 0 if otherwise).

After employing this procedure, we have found 12 months prevalence of mental and/or behavioural disorder of children and adolescents aged 11-17 years was 38.5%, whereas Irteja et al. (2020) showed 12-month prevalence rate of mental disorders of adolescents aged 13-17 years was 34.7% [43]. 

2. It would have been easier to comment on the paper if page and line numbers had been added to the text.

Author’s response: Thank you so much for this suggestion.

Page and line numbers have been added to the text.

3. Please proofread your manuscript meticulously and correct the spelling and grammatical mistakes. In parts the wording lacks accuracy

Author’s response: Thank you for your suggestions.

The manuscript has been edited by an English-speaking native, so we hope it now matches the journal standard.

4. Abstract: Please include the year of data collection in the methods section of the abstract.

Author’s response: Thanks for indicating the issue.

Year of data collection has been included in the methods section of the abstract as follows:

“Methods: Children and adolescents aged 11–17 years (n = 3152) were considered from the 2013-2014 Young Minds Matter: The Second Australian Child and Adolescent Survey of Mental Health and Wellbeing”.

5. Background: In the section of the background chapter that begins with “Therefore, instead of analyzing each socio-demographic factor separately, in this study, we have considered a latent class analysis (LCA)…”, I would describe the research gap, not the objective of the study. For example, you could write: Up to date, in these studies, socioeconomic and sociodemographic factors were only analyzed separately, not in combination.

Author’s response: Thank you for this response.

We have written a similar statement three lines before “Therefore, instead of analyzing each socio-demographic factor separately, in this study, we have considered a latent class analysis (LCA)…”,

Now we have re-written this in the following way according to your suggestions. 

“To date, in all these studies, those socioeconomic and demographic variables were only analyzed separately not in combination.”

6. In the last paragraph of the background I would recommend to write:

“Therefore, the FIRST objective of this study is to identify the cluster …”

AND: “The SECOND objective of this study is to explore the link between adolescent mental health with class membership and adjusted controlling factors.”

Author’s response: Thank you for your advice.

We have revised this section as follows-

“Therefore, the first objective of this study was to identify the cluster of the items representing socio-demographic characteristics of Australian children and adolescents aged 11-17 years by using LCA, and the second objective was to explores the link between the mental and behavioral disorders of children and adolescents with class membership and adjusted controlling factors.

7. Method: Please delete the year in brackets of reference 22. 

Author’s response: Thank you for pointing out this.

We have removed the year in brackets.

8. In the chapter “Measurements” please describe the measurement and operationalization of the sociodemographic variables. The description of the social hierarchy of Australia and the “Diagram of Australian social hierarchy and its‘characteristics” should be presented in the background chapter, not in the methods chapter. Please only provide information related to your study methodology.

Author’s response: Thank you for your nice suggestions.

Measurement and operationalization of the sociodemographic variables were clearly described in the measurement sections as follows: 

“Socio-demographic variables used in this study were represented by age (11 - 15 years vs. 15 - 18 years), Sex (male vs. female), income (less than $52,000 as low, $52,000–$129,999 as medium, and more than $130,000 as high), regional status (metropolitan vs. non-metropolitan), primary carer education (bachelor, diploma, and year-12/below), primary carer occupation (employed vs unemployed), index of relative socio-economic advantage and disadvantage (IRSAD) quintile [lowest (most disadvantaged), second, third, fourth and highest (most advantaged)] of the parents, family blending (intact family vs other families) and both parents living status in the household (yes vs no), . The index (IRSAD) summarizes data on the economic and social circumstances of residents and households in a region, including both relative advantage and disadvantage measures. Variable “income” represents the total household income per year before tax (gross income). The variables “family blending” and “both parents living status in the household” indicated the family structure of the corresponding household. Adolescents from non-intact families had a lower perception of family functioning than adolescents from intact families [3]. Families are said to be intact if both biological parents lived there and non-intact families included stepfamily, blended family, lone parent family and other family. In addition, the variable ‘both parents live in the household’ indicates what percentage of children and adolescents lived with both of their parents.”

The description of the social hierarchy of Australia and the “Diagram of Australian social hierarchy and its characteristics” have been presented in the background section.”

9. Please explain the income variable in more detail. Is the income of the household used? Was the number of household members taken into account (household equivalent income? Is it gross or net income)?

Author’s response: Thank you for your feedback.

In this study, the income variable represents the overall household income per year before tax, which indicates the gross income.

10. What is the difference between “family blending (intact family vs other families)” and “parents’ living status (whether both parents live in the household or not)”? Is it necessary to use both indicators? 

I suppose that the two variables measure almost the same thing. Do the variables refer to the biological parents? I would suggest to use the variable “family type” (“couple family” versus “one-parent family”) as Lawrence et al. (2016) (doi:10.1177/0004867415617836) does. Or alternatively: “original family (yes/no)”.

Furthermore, I am of the opinion that both variables measure family structure and not family functioning. Although it is known that there is a strong association between family structure and family functioning, they are not the same. Therefore, please move the sentence “In 2015, a study found that adolescents from non-intact households had a lower perception of family functioning, father‘s and mother‘s behavioral control, paternal psychological control, and parent-child relational traits than adolescents from intact households [25].” into the discussion.

Author’s response:Thank you for pointing out this. 

The variables “Family blending” and “Both parents living status in the household” measured family structure, and both of these variables have a significant impact on family functioning. 

In the variable” family blending”, other families include stepfamily, blended family, lone parent family and other family. Moreover, the variable “Both parents living status in the household” means the children and adolescents lived with both of their parents.

According to the ABS (Australian Bureau of Statistics) report of Australian families with children and adolescents (2011), 89% of families were “intact”, 6% were stepfamilies and 5% were blended families. In addition, 71% of all children lived with both natural parents (bio-logical parents), 4% were in stepfamilies, and 5% were in blended families. This report also stated that percentage of adolescents living with their household varied with their age. Naturally, children under the age of five were more likely to live with both of their biological parents (78%), followed by those between the ages of five to nine (72%). The least likely to be in this scenario (63%) were those between the ages of 15 to 17 years. Most of the adolescents in our study also belong to the age group of 15-17 years and a greater percentage of adolescents who were suffering from the mental and behavioral disorders also belongs to this group. For that reason, we considered both variables. 

Lawrence et.al. (2016), also used the variable “family blending” in their study but they used the name “family type’ and represented as original family, stepfamily, blended family, sole/lone parent family and other family. 

We have moved the sentence “In 2015, a study found that adolescents from non-intact households had a lower perception of family functioning, father‘s and mother‘s behavioral control, paternal psychological control, and parent-child relational traits than adolescents from intact households [34]” into the discussion section.

11. I would recommend to move the first paragraph of the chapter “Study participants and data analysis” to the end of the chapter “Data source and study design” and rename the chapter "Statistical Analysis".

Author’s response: Thank you for the suggestions.

We have moved the paragraph and change the section name according to your suggestions as-

“Statistical analysis”

12. I don’t understand why you adjusted for the variables “inability to pay gas, electricity, or telephone bills on time”, “could not pay the mortgage or rent on time”, “adults/children have gone without meals”, and “households receive a carer benefit or pension in relation to child”. I think it is an overadjustment. You may want to analyze whether the association between class membership and mental disorder is mediated by these variables. But that is not the subject of your study. I recommend to adjust only for household size.

Author’s response: Thanks for raising the issue and valuable suggestions.

According to the reviewers’ suggestions, we have adjusted the variable “household size” only and current adjusted model showed almost similar results as unadjusted model.

13. Were weights and survey procedures used in the statistical analysis? Due to the sampling (complex structure of sample), this should be done.

Author’s response: Thank you so much for this feedback.

All the estimates were weighted in the revised manuscript to represent 11-17-year-olds in the Australian population. In particular, the weighting took into consideration the overrepresentation of younger children and households with more than one child between the ages of 4 and 17 as well as the oversampling component of 16- and 17-year-olds.

14. Results: For the labeling of the age groups, a wrong mathematical symbol was used. It must be “≥15”. Alternatively, you could write “11 to 15” and “15 to 18” (see line 2 of the results chapter and Table 1).

Author’s response: Thank you.

We have changed the label according to your suggestions as follows:

“According to the age group, 51.1% of the total samples were between 15 to 18 years old and the mean age was 14.6 years.”

15. “Moreover, the study results showed that 68.1% of parents live in households.” What does this mean? 

Author’s response: Thank you so much for querying this.

According to the new analysis, this is 66.1%.

“This indicates that 66.1% of the children and adolescents lived with both of their parents”.

16. You reported a 12month prevalence of 63.5 % for mental disorders in adolescents!

In the manuscript of Lawrence et al. (2016) (doi:10.1177/0004867415617836), who used the same data, the 12month prevalence of mental disorders was 13.9 %. I recommend strongly to check the operationalization of the variable. For reasons of comparability, please build the variable analogously to the mentioned reference paper.

Author’s response: Thank you so much for pointing out this major issues. 

We have elaborately described this issue in response to the reviewer 2’s, comment# 1.

17. The associations between the single sociodemographic variables and mental disorders are not the subject of the study. Therefore, I would delete the section “Distribution of socio-demographic characteristics with adolescents’ mental health” and move Table 2 into the supplement.

After the description of the study sample, I recommend to start with the classes you found.

Author’s response: Thank you for your feedback.

We have arranged manuscript as per your suggestions and placed table 2 in the supplementary information (Table S1). In addition, we started our writing with the classes after the description of the study sample.

18. The description of the clusters could be more concise. Please highlight the main characteristics of each class.The cluster labels of class 3 “Mobile middle class” and class 4 “Growing wealthy class” cannot be deduced from the variables used in the LCA. For example, the most striking criterion of the "growing wealthy class" is that the adolescent does not live with both parents. I recommend to give the classes labels that result from the associations reported in Table 3.

Author’s response : Thank you for your feedback.

After highlighting the main characteristics of each class, we revised the description of the latent classes. Considering your recommendation, we have labelled these classes as follows-

Class-1: Lowest socio-economic status and poor family structure

Class-2: Worst education and occupational attainment and good family structure

Class-3: Average socio-economic status and good family structure.

Class-4: Good socio-economic status and worst family structure

Class-5: Highest socio-economic status and good family structure

19. In Table 4, I suggest to use the recently introduced labels for the classes and to put the class numbers in brackets.

Author’s response: Thank you so much.

We have introduced the labels according to your suggestions in table 3.

20. Following the description of the classes, I would like to know the prevalences for mental disorders stratified by class. That is the main result of your study!

Author’s response: Thank you so much for raising this issue.

We have shown this in Table 4 (please see in revised manuscript).

21. In the section “Regression modeling” I would start with the sentence “Table 5 presents the results of binary logistic regressions …”. The sentences before belong into the method chapter. 

Author’s response: Thank you for this important suggestion. 

We have revised this part according to your suggestions as follows:

“Regression modelling: Table 5 presents the results of binary logistic regressions examining the associations between identified classes and mental and behavioral disorders after adjusting for covariates. Children and adolescents belonging to class 1 (lowest socio-economic status and poor family structure) and class 4 (good socio-economic status and worst family structure) were about 2.26 and 1.62 times more likely to be suffering from mental and/or behavioral disorders compared to their class 5 (highest socio-economic status and good family structure) counterparts (OR [odd ratio]: 2.26, 95% CI [confidence interval]: 1.82-2.82 for class 1, and OR: 1.62, 95% CI: 1.26-2.08 for class 4). Class 2 (worst education and occupational attainment and good family structure) and class 3 (average socio-economic status and good family structure) also showed higher odds of mental and behavioral disorders than did class 5, but they were not statistically significant.”

Note: In the revised manuscript we have changed the reference category for better understanding of the differences among the classes and used class 5 as the reference category.

22. Regarding Table 5, 2 decimals suffice for reporting the OR and the 95 % CI.

Author’s response: Thank you. In the revised manuscript we have used two decimals in all the tables

23. The calculation of the OR only allows comparisons to the reference group. If you would calculate predictive margins, you could compare all classes with each other.

Furthermore, when the prevalence of the disease is high, the Prevalence Ratio (PR) would be the better estimator.

Author’s response: Thank you for the feedback. 

We know that, in case of analysing public health data, odds ratio is commonly used to report the strength of association between exposure and an event. The larger the odds ratio, the more likely the event is to be found with exposure. For better understanding of the of the significant difference among the classes and more precise interpretation, we have changed the reference category in our revised manuscript and used class 5 (Highest socio-economic status and good family structure) as the reference category. 

As a result, our final study outcome was described as follows:

Children and adolescents belonging to the class 1 (Lowest socio-economic status and poor family structure) and class 4 (Good socio-economic status and worst family structure) were about 2.26 and 1.62 times more likely to be suffering from mental and behavioral disorders compared to their class 5 (Highest socio-economic status and good family structure) counterparts”.

After considering your recommendations we have determined the predictive margin of the classes. And we found the following results.

 Predictive margin (95% CI)

Lowest socio-economic status and poor family structure (class1) 0.52 (0.48-0.56)

Worst education and occupational attainment and good family structure (class 2) 0.35 (0.31-0.40)

 Average socio-economic position and good family structure (class 3) 0.32 (0.29-0.36)

 Good socio-economic status and worst family structure (class 4) 0.44 (0.39-0.49)

 Highest socio-economic status and good family structure (class 5) 0.32 (0.29-0.36)

This result is similar to the distribution of the prevalence of mental and/or behavioral disorders by classes, which is shown in Table 4. We have interpreted the results in Table 4 as follows:

Table 4 shows that the prevalence of mental and/or behavioral disorders of class 1 was 52.4%. This means that, if all the respondents were from class 1, then the average prevalence rate of mental and/or behavioral disorders would have been 0.52. Class 4 shared the lowest predicted class membership (12.5%) but it had a greater prevalence (44.1%) of mental and/or behavioral disorders than did class 2 (35.2%), class 3 (32.9%), and class 5 (32.7%). As a result, it can be deduced that if all respondents had been from class 4, then the prevalence rate of mental and/or behavioral disorders would have been 0.44 on average. In addition, the second highest percentage of adolescents (23.3%) belongs to class 5, only 32.7% of whom suffered from mental and behavioral disorders”.

For that reason, we did not present this results in our manuscript. 

As we have responded in the reviewer 2’s comment# 1 that the prevalence rate of mental and behavioural disorders of children and adolescents aged 11-17 years was found to be 38.5% in our revised manuscript. Therefore, we have only estimated the odds ratio rather than prevalence ratio.

24. Discussion: The aim of your study is to analyze the association between the sociodemographic classes and adolescents’ mental disorder and not to analyzed the associations between the single sociodemographic variables and mental disorder. In particular, the results for your approach should be reported and discussed. Please revise the first paragraph of the discussion accordingly.

Author’s response: Thank you for your suggestions. We have revised the first paragraph of the discussion part according to your suggestions as follows:

“In this study LCA was used to identify distinct classes based on various socio-demographic factors of Australian children and adolescents. Patterns of socio-economic and demographic features were characterized by various levels of engagement in social behaviors across multiple domains and are differentially associated with mental and behavioral disorders [15, 22, 34, 43]. Differences in socio-demographic characteristics and mental health issues were observed in the heterogeneous classes, supporting the view that structural factors (e.g., age, gender, region, occupation, education and family blending) influence individuals’ mental health. Using nine distinct socio-demographic factors, we were able to identify 5 latent classes in our study. Mental and behavioral disorders of children and adolescents were mostly prevalent in class 1 and rarely in class 5. Although lowest percentage of children and adolescents were from class 4, prevalence of mental and behavioral disorders in this class was higher compared to class 2, class 3, and class 5. The model of binary logistic regression revealed that, in comparison to class 5, class 1 and class 4 showed higher odds of mental and behavioral disorders even after adjusting for control variable.”

25. The limitations of the study are missing completely.

Author’s response

Thank you for pointing out this error. Limitations of the study have been added in the strength and limitation sections in the revised manuscript by the following ways-

“Strengths and limitations: The strengths of this study included the use of person-centered approach (i.e., LCA) that concurrently examines the socio-demographic characteristics of children and adolescents and determined the associations between obtained latent classes and their mental and behavioral disorders by using nationally representative YMM data. This study has a few limitations that need to be mentioned. First, the 55% response rate of the survey could have created some bias that weighing did not take into consideration. Second, this study explored the association of children and adolescents’ mental and behavioral disorders with latent classes, but we did not show the relationship between latent classes with various kinds of mental and behavioral disorders in children and adolescents. Moreover, we did not consider the parents/carers’ mental health status, that may have impact on the outcome of our study. ‘Don’t know responses’ were omitted in our study, which may have a significant impact on the result of the study. Lastly, the analysis was based on a cross-sectional study and self-reported responses, where causal relationships cannot be established through a cross-sectional study, and self-reported responses compromise with the reliability and validity of measurement.”

26. Neither the discussion nor the conclusions point out the advantage of class analysis over bivariate correlation analysis. 

Author’s response

Thank you.

We have checked thoroughly and corrected the references.

Reviewers# 3 comments

1. The authors present an interesting analysis with appropriate reference to existing literature.

My major concern about the presented analysis is that the prevalence of mental disorders in this population is very high at 63.5% (Table 2). The participants were a representative sample of 11-17 year olds and based on global prevalences I would have expected the prevalence to be 10-30%. A quick look at national Australian data also suggest that this prevalence is very high. (https://www.aihw.gov.au/reports/children-youth/australias-children/contents/health/children-mental-illness ; https://www.abs.gov.au/statistics/health/mental-health)

The authors should explain why such a high prevalence was observed e.g. was this population different in some way, and if so then how generalizable are the findings.

Author’s response: Thank you so much for encouraging feedback.

Reviewers’ 2 also raised the same query. 

We have checked and reanalysed our data. In the query number 1 of reviewer 2, we have elaborately described about the measurement of the mental and behavioral disorders.

In the responses to comment 1 of reviewer 2, we also explained why rate was so high in our original submission and what changes we made in the revised version.

After revision, we found the prevalence of mental and behavioral disorders of children and adolescents aged 11-17 years was 38.5% in our study. However, a previous study (Irteja et al. (2020); https://doi.org/10.1371/journal.pone.0231180.t005) using the same data showed that the prevalence of mental disorders of adolescents aged 13-17 years was 34.7%.

2. Discussion: The authors should include a description of the strengths and weaknesses of the methods and measurements used. For example, some assumptions are made around family functioning dependent on whether the adolescent is living with their parents. There is likely to be evidence in the literature that supports this assumption but in situations where parents are arguing then the family may function better when the parents live in different households. The authors should reflect on the appropriateness of these assumptions. Also, family functioning appears to be measured using just one question around who the adolescent lives with and this warrants some reflection and critique.

Author’s response: Thank you for raising this issue and nice concluding remarks.

We have included strength and limitations in our revised version separately from discussion as follows:

“Strengths and limitations: The strengths of this study included the use of person-centered approach (i.e., LCA) that concurrently examines the socio-demographic characteristics of children and adolescents and determined the associations between obtained latent classes and their mental and behavioral disorders by using nationally representative YMM data. This study has a few limitations that need to be mentioned. First, the 55% response rate of the survey could have created some bias that weighing did not take into consideration. Second, this study explored the association of children and adolescents’ mental and behavioral disorders with latent classes, but we did not show the relationship between latent classes with various kinds of mental and behavioral disorders in children and adolescents. Moreover, we did not consider the parents/carers’ mental health status, that may have impact on the outcome of our study. ‘Don’t know responses’ were omitted in our study, which may have a significant impact on the result of the study. Lastly, the analysis was based on a cross-sectional study and self-reported responses, where causal relationships cannot be established through a cross-sectional study, and self-reported responses compromise with the reliability and validity of measurement”.

However, in our study we have considered two family structure variables (family blending, both parents living status in the household). Both factors have significant impact on family functioning. According to the ABS (Australian Bureau of Statistics) report of Australian families with children and adolescents (2011), 89% of families were “intact”, 6% were stepfamilies and 5% were blended families. In addition, 71% of all children lived with both natural parents (bio-logical parents), 4% were in stepfamilies, and 5% were in blended families. This report also stated that percentage of adolescents living with their household is varies with their age. Naturally, children under the age of five were more likely to live with both of their biological parents (78%), followed by those between the ages of five and nine (72%). The least likely to be in this scenario (63%) were those between the ages of 15-17 years. According to our study higher percentage of children and adolescents aged 15-17 years were suffering from mental and behavioral disorders. 

3. A main conclusion is that improvement of mental health for children and adolescents requires improved socioeconomic status of the parents and family structure. 

The potential importance of the mental health of the parents should be discussed as poor parental mental health could be one of the reasons that the family have poor socioeconomic status and/or why the family structure is disturbed.

Author’s response: Thank you.

According to the reviewers’ suggestions we have revised the conclusions of the abstract as follows.

“Abstract: 

Conclusions: Among the five latent classes, children and adolescents from classes 1 and 4 have a higher risk of developing mental and behavioral disorders. The findings suggest that the improvement of mental health for children and adolescents requires improving their family structure and the socioeconomic status of the parents. 

We have stated in our strength and limitation section that we did not consider parental or carer mental health in this study.

4. Conclusions

Please elaborate on recommendation to screen for patterns of socio-demographic factors to detect and prevent mental disorders. I presume that the authors are not suggesting that mental health or social services should be provided only to certain classes but perhaps they think that some targeting might be possible? 

Author’s response: We thank the reviewer for the careful consideration and constructive feedback. 

In this study, our target was not only to suggest particular classes for improving mental and behavioral disorders of children and adolescents but also to recommend the corresponding factors that have significant influence on the prevalence of mental and behavioral disorders.

We have revised the conclusions in the following ways:

“The present study identified five classes based on various socio-demographic characteristics and highlighted the association of mental and behavioral disorders of Australian children and adolescents. Class 1 (Lowest socio-economic status and poor family structure) and class 4 (Good socio-economic status and worst family structure) showed higher odds of mental and behavioral disorders among children and adolescents than class 5 (Highest socio-economic status and good family structure). In addition, compared to classes 2, 3, and 5, children and adolescents from class 4 were more likely to have suffered from mental and behavioral disorders, although this class shared the lowest class membership and consisted good socio-economic status. This indicates that, improving the socioeconomic position of the parents as well as the family structures is necessary for improving the mental health of children and adolescents. The associations between the pre-identified clusters and mental health highlighted the importance of screening the patterns of socio-demographic factors as a promising step to detect and prevent mental and behavioral disorders among at-risk children and adolescents more effectively.

5. Minor comments

Abstract: The sentence ‘However, no study has yet been conducted on a model-based cluster analysis of socio-demographic characteristics with mental health’ could be rephrased to indicate that the authors could not find evidence of such a published study.

Author’s response: Thank you for your recommendations.

We have rephrased this in the following way-

“However, no research has been found on a model-based cluster analysis of socio-demographic characteristics with mental health.”

6. It is possible (likely?) that someone has tried this at some point somewhere.

Author’s response: A study has been published in August 2022 (https://doi.org/10.1371/journal.pone.0272614) by Daundasekara et al. where the authors used latent class analysis to identify socio-economic and health risk profiles among mothers of young children predicting longitudinal risk of food insecurity [19]. However, that study is not in the Australian contexts, and it is based on US data. Importantly, the scope of our study is distinctly different from that of Daundasekara and colleagues.

We have added this reference in our background section in the revised manuscript as follows:

“Daundasekara et. al., (2022) used LCA to identify socio-economic and health risk profiles among mothers of young children predicting longitudinal risk of food insecurity”.

7. Presenting data to 2 decimal points is usually sufficient and easier for the reader to digest.

Author’s response: Thank you for the suggestions.

In the revised version we have used two decimal points.

8. Background: Line 1- are mental health issues rising or is their detection rising?

Author’s response: Thank you for correcting this.

We have corrected this line in the following way-

“Both incidence and prevalence of mental health conditions are rising significantly among young Australians.”

9. Measurements 

Para 2

ADHD and conduct disorder are behavioral disorders so a more correct description would be ‘mental and behavioral disorders’

Author’s response: Thank you for indicating this issue.

We have replaced the word “mental disorders” by “mental and behavioral disorders” in the revised version and accordingly we have changed the title of the manuscript as:

“A latent class analysis of the socio-demographic factors and mental and behavioral disorders of Australian children and adolescents”

10. Fig 2, The word ‘regions’ should be explained as has not been defined previously. In Table 1 this is defined but then replaced with ‘remoteness’ in Table 4. I recommend the use of consistent terms

Author’s response: Thank you for pointing out this error-

In the revised version of the manuscript, we have used the label “regional status” in all the tables. 

11. Family blending

Intact families are described as those families in which their parents are married and living together

Other families include step, blended and single-parent families.

If an adolescent is living with both parents but they are not married, which category do they fall into?

Author’s response: Thank you for your feedback.

In this study we have considered both these variables (family blending, and both parents living in the household) as independent family structure variables. Both have significant association with mental and behavioral disorders of children and adolescents. 

In the YMM data dictionary, there is no clear indication whether both parents are married or not as we know that non-marital cohabitation is socially and culturally acceptable in the Western World. The question asked- “Both parents living in household?” and the responses was “yes” or “no”.

12. Table 2 Has both row % (mental disorder) and column % (total). This can be a bit confusing for the reader so I suggest you reformat and/or label the % as row and column where appropriate.

Author’s response: Thank you for this suggestion.

We have used the column percentage in the revised version in Table 1 and Table S1. In table 4, we used row percentage in mental and behavioral disorders and column percentage in predicted class membership.

---

## [Decision Letter · Decision Letter 1]

3 Apr 2023

PONE-D-22-28764R1A latent class analysis of the socio-demographic factors and mental and behavioral disorders of Australian children and adolescents.PLOS ONE

Dear Dr. Afroz,

Thank you for submitting your manuscript to PLOS ONE. After careful consideration, we feel that it has merit but does not fully meet PLOS ONE’s publication criteria as it currently stands. Therefore, we invite you to submit a revised version of the manuscript that addresses the points raised during the review process.

The revised version should address the remaining issues. Please submit your revised manuscript by May 18 2023 11:59PM. If you will need more time than this to complete your revisions, please reply to this message or contact the journal office at plosone@plos.org. Please include the following items when submitting your revised manuscript:A rebuttal letter that responds to each point raised by the academic editor and reviewer(s). You should upload this letter as a separate file labeled 'Response to Reviewers'.A marked-up copy of your manuscript that highlights changes made to the original version. You should upload this as a separate file labeled 'Revised Manuscript with Track Changes'.An unmarked version of your revised paper without tracked changes. You should upload this as a separate file labeled 'Manuscript'.If applicable, we recommend that you deposit your laboratory protocols in protocols.io to enhance the reproducibility of your results. Protocols.io assigns your protocol its own identifier (DOI) so that it can be cited independently in the future. For instructions see: https://journals.plos.org/plosone/s/submission-guidelines#loc-laboratory-protocols. Additionally, PLOS ONE offers an option for publishing peer-reviewed Lab Protocol articles, which describe protocols hosted on protocols.io. Read more information on sharing protocols at https://plos.org/protocols?utm_medium=editorial-emailutm_source=authorlettersutm_campaign=protocols.

We look forward to receiving your revised manuscript.

Kind regards,

Petri Böckerman

Academic Editor

PLOS ONE

Journal Requirements:

Additional Editor Comments (if provided):

The revised version should address the remaining issues.

Reviewers' comments:

Reviewer's Responses to Questions

**Comments to the Author**

1. If the authors have adequately addressed your comments raised in a previous round of review and you feel that this manuscript is now acceptable for publication, you may indicate that here to bypass the “Comments to the Author” section, enter your conflict of interest statement in the “Confidential to Editor” section, and submit your "Accept" recommendation.

Reviewer #1: (No Response)

Reviewer #2: (No Response)

2. Is the manuscript technically sound, and do the data support the conclusions?

Reviewer #1: Yes

Reviewer #2: Partly

3. Has the statistical analysis been performed appropriately and rigorously? 

Reviewer #1: Yes

Reviewer #2: Yes

4. Have the authors made all data underlying the findings in their manuscript fully available?

Reviewer #1: Yes

Reviewer #2: No

5. Is the manuscript presented in an intelligible fashion and written in standard English?

Reviewer #1: Yes

Reviewer #2: No

6. Review Comments to the Author

Reviewer #1: Authors have addressed most of the issues I have raised.

I have some minor comments:

1) Literature review is still rather short. The following sentences could be presented in more detail and include more references to previous studies: "Various socio-demographic characteristics such as age, regional status, gender, parental education, parental employment status, and household income can predict mental health and aggressive behaviors in children and adolescents [9–11]. The mental health of children and adolescents is also expressively influenced by their family structure [12]."

2) Thank you for including policy and practice recommendations. Potential mechanisms explaining the associations are still missing.

3) Potential limitations related to the use of LCA needs to be discussed

Reviewer #2: Dear authors,

Thank you for revising the manuscript. The manuscript has now improved significantly. However, I still have a few suggestions which should be addressed.

1.) Now a prevalence of mental and behavioral disorders of 38.5% is reported, which is significantly lower than the prevalence in the previous version. However, it is still unclear how mental and behavioral disorder was operationalized and why the prevalence is significantly higher than in other studies. The literature cited does not provide any information on this. This also applies to the study by Irteja et al., which is a self-citation of a co-author. I suppose that you do not consider any functional impairments in your operationalization.

More than a third of the 11- to 17-year-olds are labeled as mentally or behavioral disordered. What is the rationale of operationalizing mental disorders in this way? You refer in your first sentence to the WHO reporting a prevalence of 14% of 10-19-year-olds suffering from mental health difficulties. Lawrence et al. (2016) (https://doi.org/10.1177/00048674156178) reported for Australia a prevalence of 12.8 % in female adolescents and 15.9 % in male adolescents aged 12 to 17.

An important argument for using the standardized operationalization considering functional impairments is that, given this high prevalence, calculating a logistic regression is controversial.

Lines 311-313: The sentence “According to Irteja et al. (2020), 12-months prevalence rate of mental disorders among adolescents aged 13-17 years old was 34.7% [43].” should be moved to the chapters background or discussion.

In Table 4 it is sufficient to report only the prevalence (yes). The other two columns can be deleted.

2.) The operationalization of the variables “family blending” and “both parents living status in the household” is still not clearly described in the method chapter (lines 234, 235). It is also still unclear what’s the difference between both variables.

From my point of view, both family structure variables are very similar, so that the family structure has the double weight in the LCA. This in turn leads to the strong division of the classes regarding the family structure.

Additionally, the description of family types that do not consist of both biological parents is stigmatizing. The family structure should not be labeled as “good”, “bad” or “worst”. You could describe the family structure in a more objective way, e.g.: "The cluster is characterized by a very high proportion of non-intact families." or: “Class 4 - High socioeconomic status and non-intact family structure”.

Furthermore, the aim of health promotion and prevention is not to improve the family structure (lines 51,52 and 505), but to improve the mental health of children and adolescents living in non-intact families. From my point of view, the stigmatizing description regarding some family structures needs to be revised throughout the manuscript.

And please move the sentence “Adolescents from non-intact families had a lower perception of family functioning than did adolescents from intact families [34].” (lines 235-237) from the method chapter into the chapters background or discussion.

Lines 95-99: What is about the single-parent families? Are they part of the intact families?

Lines 302-303: How can it be that 66.1% live with both parents, but only 61.8% are considered to be living in an intact family?

Line 450: Non-intact family structures do not “consequently” possess weak family functioning. But: Referring to other studies, it can be assumed that a non-intact family structure is associated with a weak family function.

3.) I would suggest changing the manuscript title as follows: A latent class analysis of the socio-demographic factors and associations with mental and behavioral disorders among Australian children and adolescents

Best regards,

Petra Rattay

7. PLOS authors have the option to publish the peer review history of their article (what does this mean?). If published, this will include your full peer review and any attached files.

Reviewer #1: No

Reviewer #2: **Yes: **Petra Rattay

---

## [Author Response · Author response to Decision Letter 1]

15 Apr 2023

A latent class analysis of the socio-demographic factors and associations with mental and behavioral disorders among Australian children and adolescents

Dear Editor,

Thank you very much for providing us with the editor and reviewer’s feedback and the opportunity to revise the manuscript. We appreciate the editor and the reviewers’ insightful comments and suggestions. We have revised the paper as per the editor and reviewer’s comments and a point-by-point response to the comments is provided below: 

Best regards,

Authors

Reviewer #1 comments and author responses

Authors have addressed most of the issues I have raised.

I have some minor comments:

Comment 1: Literature review is still rather short. The following sentences could be presented in more detail and include more references to previous studies: "Various socio-demographic characteristics such as age, regional status, gender, parental education, parental employment status, and household income can predict mental health and aggressive behaviors in children and adolescents [9–11]. The mental health of children and adolescents is also expressively influenced by their family structure [12]."

Authors’ responses

Thank you very much for your comments. We have revised those sentences with more information by presenting them as follows:

According to the Mission Australia’s latest Youth Survey Report 2022, 33.9% of young people viewed mental health as a significant national concern. Almost three in 10 (28.8%) young people reported experiencing severe psychological distress, and nearly one-quarter (23.5%) stated feeling lonely most of the time. Little more than half of individuals (53.4%) have at some stage in their lifetimes required mental health care. In 2020, 49.9% of young people were optimistic about their future, but this proportion has steadily decreased since then [7]. [Page:4; Lines:81-86]

For instance, household income and parental education had a greater impact on children’s and adolescents’ mental health issues than did parental unemployment or poor occupational status, which refers a low position in the occupational hierarchy [14]. Moreover, children of parents with university degrees are more likely to have greater levels of psychological well-being than children of parents without a university degree [15]. [Page: 5; Lines: 100-105]

Numerous studies [20], [21] have shown that children from non-intact families are more likely to experience negative psychological consequences than children from intact families. [Page: 4; Lines:116-118]

Comment 2. Thank you for including policy and practice recommendations. Potential mechanisms explaining the associations are still missing.

Authors’ response

Thank you so much.

The generalized linear model with the log link binomial family (log-binomial regression model) was taken into consideration in the revised manuscript due to the high prevalence of mental and behavioral disorders [48] and determined the prevalence ratio instead of the odds ratio (see Table 5 in the manuscript) [Page:21]

Comment 3: Potential limitations related to the use of LCA needs to be discussed.

Authors’ response

Thank you so much for raising this issue. We have added the limitations related to the use of LCA in the section strength and limitations as follows:

“Third, LCA allocated individuals to classes based on their likelihood of belonging to a class given their patterns of indicator variable scores. However, there is no certainty that the class assignments were completed accurately”. [Page:24, Lines: 500-503]

Reviewer# 2 comments and author responses

Dear authors, 

Thank you for revising the manuscript. The manuscript has now improved significantly. However, I still have a few suggestions which should be addressed.

Comment 1. Now a prevalence of mental and behavioral disorders of 38.5% is reported, which is significantly lower than the prevalence in the previous version. However, it is still unclear how mental and behavioral disorder was operationalized and why the prevalence is significantly higher than in other studies. The literature cited does not provide any information on this. This also applies to the study by Irteja et al., which is a self-citation of a co-author. I suppose that you do not consider any functional impairments in your operationalization.

More than a third of the 11- to 17-year-olds are labeled as mentally or behavioral disordered. What is the rationale of operationalizing mental disorders in this way? You refer in your first sentence to the WHO reporting a prevalence of 14% of 10-19-year-olds suffering from mental health difficulties. Lawrence et al. (2016) (https://doi.org/10.1177/00048674156178) reported for Australia a prevalence of 12.8 % in female adolescents and 15.9 % in male adolescents aged 12 to 17.

An important argument for using the standardized operationalization considering functional impairments is that, given this high prevalence, calculating a logistic regression is controversial.

Lines 311-313: The sentence “According to Irteja et al. (2020), 12-months prevalence rate of mental disorders among adolescents aged 13-17 years old was 34.7% [43].” should be moved to the chapters background or discussion.

In Table 4 it is sufficient to report only the prevalence (yes). The other two columns can be deleted.

Authors’ response

Thank you for your feedback.

In this study, our outcome variable is “mental and/or behavioral disorders”. We have included children as young as 11 and investigated the age range of 11 to17 years old. Several questions in the survey were used to indicate the presence of mental and/or behavioral problems or distress that despite not meeting full diagnostic criteria, may be of clinical significance, or be of concern to parents, carer givers, or children and adolescents. This study considered all types of anxiety disorders, major depressive disorder, ADHD and conduct disorder as mental and/or behavioral disorders. Each of the mental and/or behavioral diseases was measured using the corresponding criteria, which considered both their major and minor symptoms. 

For example, DSM-5 criteria were used to measure conduct disorder (a behavioral problem affecting children and adolescents). If at least four symptoms were present, then its response was “yes”. DSM‐IV criteria for major depressive disorder require the presence of at least six symptoms. Social phobia (SP), separation anxiety disorder (SED), generalized anxiety disorder (GAD), and obsessive-compulsive disorder (OCD) are the types of anxiety disorder (AnxD). All mental and behavioral disorders included in AnxD were measured separately. For example, to measure GAD, anxiety and worry were accompanied by at least three physical or cognitive symptoms. After measuring SP, SED, GAD and OCD, we measured AnxD: whether the children and adolescents have had any of these four types of anxiety disorder. Response included ‘Yes’ and coded as 1 if a child has at least one of these anxiety disorders. ADHD is a behavior disorder, usually begins in childhood but may continue into adulthood. It is the most commonly diagnosed behavioral disorder in children. For ADHD, symptoms were divided into two categories: inattention (A1) and hyperactivity (A2). DSM-IV criteria were used to measure A1 and A2 separately. If both criteria A1 and A2 were met, then response was “yes” (Attention-Deficit/Hyperactivity Disorder). In our study we have also considered the response as “yes” for ADHD when criterion A1 was met but criterion A2 was not met; this type of ADHD is referred to as ADHD predominantly inattentive type [1]. Similarly, the response was deemed as “yes” when criterion A2 was met but criterion A1 was not met and it is called ADHD predominantly hyperactive-impulsive type. Attention-Deficit/Hyperactivity Disorder was there after coded as 1 when either or both criteria were met. Finally, we added a variable ‘mental and/or behavioral disorder’. If a child has at least one of the mental or behavioral issues, the responses was “Yes”. 

The prevalence of mental or behavioral disorders is comparatively higher in our study because we accounted for all the major and minor symptoms associated with each of them. In addition, we have considered the partial requirements for confirming the presence of the corresponding behavioral problems (such as ADHD). The main reason for doing that was to determine the class-wise prevalence of mental and behavioral disorders in children and adolescents based on their initial symptoms in order to facilitate early diagnosis. According to our study objectives, we intended to investigate the relationship between mental and behavioral disorders of children and adolescents and various latent classes in order to distinguish vulnerable classes from privileged classes. As is well known, some mental and/or behavioral disorders are the most frequently diagnosed disorders in children and adolescents, therefore, it is important to ascertain their prevalence by class at the earliest possible stage in order to facilitate early diagnosis and treatment. This might help reduce a country's annual health care cost burden.

According to previous studies (e.g., Lawrence et al. 2015), the prevalence of mental disorders is quite low because they did not consider partial diagnostic criteria to confirm the presence of corresponding disorder. However, we believe that the prevalence of mental disorder is actually much lower, although the Mission Australia’s new Youth Survey Report 2022 revealed an alarming information about the mental health of young Australian. According to this report, 33.9% of young people believed mental health to be a significant national concern. Almost three in 10 (28.8%) young people reported experiencing severe psychological distress, and nearly one in four (23.5%) said they felt lonely most of the time. About half (53.4%) have at some point in their lives needed assistance with their mental health. While half (49.9%) of young people were positive about their futures, this percentage has increasingly declined since 2020. [Page:4; Lines:81-86]

We estimated the prevalence ratio using the log link binomial family in the generalized linear model (log binomial regression model), as per your recommendation and the reference [48] due to the high prevalence of mental and behavioral disorder, rather than the odds ratio of logistic regression. [Page:21]

Table 5 Log-binomial regression model of predicting latent class membership (unadjusted and adjusted) and control variables. 

Latent classes (ref. Unadjusted Adjusteda

Highest socio-economic status and 

intact family structure (Class 5)) Coefficient PR(95% CI of PR) Coefficient PR(95% CI of PR)

Good socio-economic status and 

non-intact family structure (Class 4) 0.30*** 1.35 (1.16, 1.57) 0.29*** 1.35 (1.15, 1.56)

Average socio-economic status and 0.01 1.01(0.87, 1.15) 0.01 1.01 (0.88, 1.15)

intact family structure (Class 3)

Worst education and occupational 0.07 1.08 (0.91, 1.27) 0.08 1.09 (0.92, 1.28)

Attainment and intact family

 structure (Class 2)

Lowest socio-economic status and 0.47*** 1.60(1.41, 1.82) 0.47*** 1.60(1.41, 1.82)

non-intact family structure (Class 1)

Note: PR-Prevalence ratio; CI-Confidence interval; *: p0.05; **: p0.01; ***: p0.001

 a Control variable: “Household size”.

Comment 2: Lines 311-313: The sentence “According to Irteja et al. (2020), 12-months prevalence rate of mental disorders among adolescents aged 13-17 years old was 34.7% [43].” should be moved to the chapters background or discussion.

In Table 4 it is sufficient to report only the prevalence (yes). The other two columns can be deleted.

Authors’ response

According to the reviewer’s suggestions, the sentence “According to Irteja et al. (2020), 12-months prevalence rate of mental disorders among adolescents aged 13-17 years old was 34.7% [10] has been moved to the background section. [Page:4; Lines:89-91]

In addition, reporting only the prevalence other two columns have been deleted from Table 4. (Page:20)

Comment 3: The operationalization of the variables “family blending” and “both parents living status in the household” is still not clearly described in the method chapter (lines 234, 235). It is also still unclear what’s the difference between both variables.

From my point of view, both family structure variables are very similar, so that the family structure has the double weight in the LCA. This in turn leads to the strong division of the classes regarding the family structure.

Additionally, the description of family types that do not consist of both biological parents is stigmatizing. The family structure should not be labeled as “good”, “bad” or “worst”. You could describe the family structure in a more objective way, e.g.: "The cluster is characterized by a very high proportion of non-intact families." or: “Class 4 - High socioeconomic status and non-intact family structure”.

Furthermore, the aim of health promotion and prevention is not to improve the family structure (lines 51,52 and 505), but to improve the mental health of children and adolescents living in non-intact families. From my point of view, the stigmatizing description regarding some family structures needs to be revised throughout the manuscript.

And please move the sentence “Adolescents from non-intact families had a lower perception of family functioning than did adolescents from intact families [34].” (lines 235-237) from the method chapter into the chapters background or discussion.

Lines 95-99: What is about the single-parent families? Are they part of the intact families?

Lines 302-303: How can it be that 66.1% live with both parents, but only 61.8% are considered to be living in an intact family?

Line 450: Non-intact family structures do not “consequently” possess weak family functioning. But: Referring to other studies, it can be assumed that a non-intact family structure is associated with a weak family function.

Authors’ responses:

Thank you for your comments and suggestions.

In this study, the variable “family blending” contained the categories intact family and other families (stepfamily, blended family, and lone parent family). Families are said to be intact if both biological parents lived together, and according to ABS [18], an intact family is a couple family with at least one child who is the natural or adopted child of both partners in the couple, and no step children. 

The variable “Both parents living status in the household” in our study refers to the children and adolescents who reside with both parents. However, in the YMM data dictionary there is no clear indication of whether or not both parents were biological. Others couple families, such as blended or step couple, may be included here. For whole sample (n=6310) in the YMM data (parent), we have found that 73.8% children and adolescents aged 4 to 17 years lived with both parents and 68.6% belonged to intact families. In our study, about 66.1% of children and adolescents aged 11-17 years lived with both parents and 61.3% were members of intact families. We included the variable "both parents living status in the household" as a family structure variable in our study to examine the variation in mental health issues among adolescents who lived with both parents versus those who do not, where both parents belong to any form of couple families (including step-family and blended family). The family structure variable “family blending” was also used in this study, where adolescents were either from intact families or non-intact families. Intact families refer to families in which both biological parents are present in the household. Both variables contribute independently to the formation of a healthy family structure. For that reason, we have included both variables in our study. Consequently, we believe it is impractical to double the weight in the LCA while forming clusters. This is evidenced by the similarity of the clustering patterns when one variable is removed.

According to the reviewers’ suggestions, we have renamed the classes as follows:

Class-1: Lowest socio-economic status and non-intact family structure

Class-2: Worst education and occupational attainment and intact family structure

Class-3: Average socio-economic status and intact family structure.

Class-4: Good socio-economic status and non-intact family structure

Class-5: Highest socio-economic status and intact family structure

We have revised the description regarding family structure throughout the manuscript. 

# As per your suggestion, we have moved the sentence “Adolescents from non-intact families had a lower perception of family functioning than did adolescents from intact families [19]” from method section to the background section. [Page:5; Lines:115-116]

# In YMM data, single-parent families are the part of non-intact families.

# In an intact family, parents should be biological. However, in the variable “both parents living status in the household”, all couple family’s parents (may or may not biological parents) were included here. It might be the reason that 66.1% of children and adolescents lived with both parents, however, only 61.8% were from intact families.

# In our previous manuscript we have written that classes 1 and 4 have very poor family structures and consequently possess weak family functioning. According to past research, good family functioning has a positive impact on the mental health of children and adolescents.

In the revised manuscript, we have written this as follows:

“Children and adolescents in classes 1 and 4 have non-intact family structures and consequently possess weak family functioning. According to past research, good family functioning has a positive impact on the mental health of children and adolescents.” [Pages:22-23; Lines: 459-461]

Comment 4: I would suggest changing the manuscript title as follows: A latent class analysis of the socio-demographic factors and associations with mental and behavioral disorders among Australian children and adolescents.

Authors’ response

According to your suggestion we have changed the manuscript title as follows:

“A latent class analysis of the socio-demographic factors and associations with mental and behavioral disorders among Australian children and adolescents”

 …………………………….

Additional reference:

[1] S. A. and M. H. S. A. SAMSHA, “DSM-5 Changes: Implications for Child Serious Emotional Disturbance (unpublished internal documentation),” no. June, p. 66, 2016, [Online]. Available: www.ncbi.nlm.nih.gov/books/NBK519712/table/ch3.t8/

---

## [Decision Letter · Decision Letter 2]

2 May 2023

PONE-D-22-28764R2A latent class analysis of the socio-demographic factors and associations with mental and behavioral disorders among Australian children and adolescentsPLOS ONE

Dear Dr. Afroz,

Thank you for submitting your manuscript to PLOS ONE. After careful consideration, we feel that it has merit but does not fully meet PLOS ONE’s publication criteria as it currently stands. Therefore, we invite you to submit a revised version of the manuscript that addresses the points raised during the review process. The revised version should address the remaining concerns.

We look forward to receiving your revised manuscript.

Kind regards,

Petri Böckerman

Academic Editor

PLOS ONE

Journal Requirements:

Additional Editor Comments:

The revised version should address the remaining concerns.

Reviewers' comments:

Reviewer's Responses to Questions

**Comments to the Author**

1. If the authors have adequately addressed your comments raised in a previous round of review and you feel that this manuscript is now acceptable for publication, you may indicate that here to bypass the “Comments to the Author” section, enter your conflict of interest statement in the “Confidential to Editor” section, and submit your "Accept" recommendation.

Reviewer #1: All comments have been addressed

Reviewer #2: (No Response)

2. Is the manuscript technically sound, and do the data support the conclusions?

Reviewer #1: Yes

Reviewer #2: Yes

3. Has the statistical analysis been performed appropriately and rigorously? 

Reviewer #1: Yes

Reviewer #2: Yes

4. Have the authors made all data underlying the findings in their manuscript fully available?

Reviewer #1: (No Response)

Reviewer #2: No

5. Is the manuscript presented in an intelligible fashion and written in standard English?

Reviewer #1: (No Response)

Reviewer #2: No

6. Review Comments to the Author

Reviewer #1: Thanks for addressing all the issues. I have no further comments. I recommend manuscript to be accepted for publication in Plos One.

Reviewer #2: Dear authors,

Thank you for revising the manuscript a second time. The manuscript has now improved again significantly.

However, I still have concerns about the operationalization of the variable "mental and behavioral disorders" and a resulting overdiagnosis of mental and behavioral disorders among adolescents. Of course, in terms of freedom of research, you can operationalize "mental and behavioral disorders" as you see fit. In my opinion the operationalization as well as the problem of overdiagnosis should be discussed critically in the limitation section.

Therefore, I would like to ask you to include the criticism regarding an overdiagnosis of mental and behavioral disorders among adolescents as well as the reasoning from your answer letter (see text below) to the limitations chapter in the manuscript (perhaps slightly shortened or adapted). Then – despite my concerns – I agree with the operationalization of the variable.

- The part from your answer letter: “The prevalence of mental or behavioral disorders is comparatively higher in our study because we accounted for all the major and minor symptoms associated with each of them. In addition, we have considered the partial requirements for confirming the presence of the corresponding behavioral problems (such as ADHD). The main reason for doing that was to determine the class-wise prevalence of mental and behavioral disorders in children and adolescents based on their initial symptoms in order to facilitate early diagnosis. According to our study objectives, we intended to investigate the relationship between mental and behavioral disorders of children and adolescents and various latent classes in order to distinguish vulnerable classes from privileged classes. As is well known, some mental and/or behavioral disorders are the most frequently diagnosed disorders in children and adolescents, therefore, it is important to ascertain their prevalence by class at the earliest possible stage in order to facilitate early diagnosis and treatment.”

Secondly, regarding the family structure, I still find your wording partially imprecise or even discriminating.

Lines 53-55 (in the clear copy version): “The findings suggest that the improvement of mental health for children and adolescents requires improving their family structure and the socioeconomic status of the parents.”

I suggest rephrasing the sentence as follows:

- The findings suggest that health promotion and prevention as well as combating poverty are needed to improve the mental health in particular among children and adolescents living in non-intact families and in families with a low socio-economic status.

Please also rephrase the sentence in lines 482-484 and 517-518, accordingly. I added a suggestion at the end.

In the methods chapter, you refer to ASB's definition that single-parent families are not considered to be intact families. Therefore, please check the following sentence again and consider the single-parent families:

Lines 108-110: "According to the Australian Bureau of Statistics (ABS) report (2013) on Australian families with children and adolescents, 89% of families were “intact”, 6% were stepfamilies and 5% were blended families [18]."

Lines 455-457: "The main disadvantage of this class was that no children and adolescents had come from intact families, and no one lived with either of their parents, thereby indicating relatively poor family functioning."

I suggest rephrasing the sentence as follows:

- The main disadvantage of this class was that no children and adolescents had come from intact families, and no one lived with both parents, thereby indicating relatively poor family functioning.

Lines 459-461: "In our study, we found that children and adolescents in classes 1 and 4 have very poor non-intact family structures and consequently possess weak family functioning."

I suggest rephrasing the sentence as follows:

- In our study, we found that most children and adolescents in classes 1 and 4 lived in non-intact families which is more often associated with poorer family functioning.

Could you please clarify this sentence? I don't understand what is meant here.

Lines 473-475: "The ABS report on Australian families with children and adolescents (2011) showed that the lowest percentage of adolescents 15–17 years old lived with both of their natural parents [13]."

With regard to the conclusions, I suggest not to re-present the results again but to move the sentences from lines 482-490 into this chapter.

Suggestion for the conclusions chapter:

- "The present study has some important public health implications as it has identified the cluster of items that represent socio-demographic characteristics and revealed their associations with the mental and behavioral disorders of children and adolescents. The significant impact of individual class on the mental and behavioral disorders will help government and non-government agencies as well as practitioners differentiate the vulnerable classes from privileged classes in formulating health-related policy and strategy." The findings indicate that health promotion and prevention as well as combating poverty are needed to improve the mental health in particular among children and adolescents living in non-intact families and in families with a low socio-economic status.

After this revision, I agree to the publication of the manuscript.

Best regards,

Petra Rattay

7. PLOS authors have the option to publish the peer review history of their article (what does this mean?). If published, this will include your full peer review and any attached files.

Reviewer #1: No

Reviewer #2: **Yes: **Petra Rattay

---

## [Author Response · Author response to Decision Letter 2]

3 May 2023

May 4, 2023

A latent class analysis of the socio-demographic factors and associations with mental and behavioral disorders among Australian children and adolescents

Dear Editor,

Thank you very much for providing us with the academic editor and reviewer’s feedback and the opportunity to revise the manuscript. We appreciate the editor and the reviewers’ insightful comments and suggestions. We have revised the paper as per the editor and reviewer’s comments and a point-by-point response to the comments is provided below: 

Best regards,

Authors

Reviewer #1 comments and author responses

Comment: Thanks for addressing all the issues. I have no further comments. I recommend manuscript to be accepted for publication in Plos One.

Authors’ responses

Thank you very much for your comments. 

Reviewer #2 comments and author responses

Comment 1: Dear authors, Thank you for revising the manuscript a second time. The manuscript has now improved again significantly.

However, I still have concerns about the operationalization of the variable "mental and behavioral disorders" and a resulting overdiagnosis of mental and behavioral disorders among adolescents. Of course, in terms of freedom of research, you can operationalize "mental and behavioral disorders" as you see fit. In my opinion the operationalization as well as the problem of overdiagnosis should be discussed critically in the limitation section.

Therefore, I would like to ask you to include the criticism regarding an overdiagnosis of mental and behavioral disorders among adolescents as well as the reasoning from your answer letter (see text below) to the limitations chapter in the manuscript (perhaps slightly shortened or adapted). Then – despite my concerns – I agree with the operationalization of the variable.

- The part from your answer letter: “The prevalence of mental or behavioral disorders is comparatively higher in our study because we accounted for all the major and minor symptoms associated with each of them. In addition, we have considered the partial requirements for confirming the presence of the corresponding behavioral problems (such as ADHD). The main reason for doing that was to determine the class-wise prevalence of mental and behavioral disorders in children and adolescents based on their initial symptoms in order to facilitate early diagnosis. According to our study objectives, we intended to investigate the relationship between mental and behavioral disorders of children and adolescents and various latent classes in order to distinguish vulnerable classes from privileged classes. As is well known, some mental and/or behavioral disorders are the most frequently diagnosed disorders in children and adolescents, therefore, it is important to ascertain their prevalence by class at the earliest possible stage in order to facilitate early diagnosis and treatment.”

Authors’ responses

Thank you for your comment. 

We have included the following criticism regarding the overdiagnosis of mental and behavioral disorders in adolescents, as well as the reasoning in the strength and limitation sections:

“Second, the prevalence of mental or behavioral disorders is comparatively higher in our study because we considered the partial criteria for confirming the existence of specific mental and/or behavioral problems (such as ADHD) by taking into consideration all of their symptoms. The main reason for doing that was to determine the class-wise prevalence of mental and/or behavioral disorders in children and adolescents based on their initial symptoms to facilitate early diagnosis. According to our study objectives, we intended to investigate the relationship between mental and behavioral disorders of children and adolescents and various latent classes in order to distinguish vulnerable classes from privileged classes. As is widely known, some mental and/or behavioral disorders are the most commonly diagnosed disorders in children and adolescents, therefore, it is important to ascertain their prevalence by class at the earliest possible stage to facilitate early diagnosis and treatment.” However, over-diagnosing mental issues might lead to mis-medication and social stigmas.” [page:24; lines: 495-507]

Comment 2: Secondly, regarding the family structure, I still find your wording partially imprecise or even discriminating.

Lines 53-55 (in the clear copy version): “The findings suggest that the improvement of mental health for children and adolescents requires improving their family structure and the socioeconomic status of the parents.”

I suggest rephrasing the sentence as follows:

- The findings suggest that health promotion and prevention as well as combating poverty are needed to improve the mental health in particular among children and adolescents living in non-intact families and in families with a low socio-economic status.

Please also rephrase the sentence in lines 482-484 and 517-518, accordingly. I added a suggestion at the end.

Authors’ responses:

Thank you so much for your suggestions.

We have revised the conclusion of the abstract as per your suggestions as follows:

“Conclusions: Among the five latent classes, children and adolescents from classes 1 and 4 have a higher risk of developing mental and behavioral disorders. The findings suggest that health promotion and prevention as well as combating poverty are needed to improve the mental health in particular among children and adolescents living in non-intact families and in families with a low socio-economic status.” [page: 03; lines: 53-56]

We rephrased the sentences in the mentioned lines (482-484) in discussion and (517-519) in conclusions, accordingly as per your recommendations as follows. 

“To enhance mental health, particularly among children and adolescents living in non-intact families and in households with low socioeconomic status, health promotion and prevention as well as tackling poverty are necessary.” [pages:23-24; lines:485-488]

“The findings indicate that health promotion and prevention as well as combating poverty are needed to improve mental health among children and adolescents living in non-intact families and in families with a low socio-economic status.” [pages:25; lines:524-526] 

Comment 3: In the methods chapter, you refer to ASB's definition that single-parent families are not considered to be intact families. Therefore, please check the following sentence again and consider the single-parent families:

Lines 108-110: "According to the Australian Bureau of Statistics (ABS) report (2013) on Australian families with children and adolescents, 89% of families were “intact”, 6% were stepfamilies and 5% were blended families [18]."

Authors’ responses: 

Thank you for your comment. 

Lines 108-110 contain information regarding couple families only and we presented this according to the ABS report.

However, according to your suggestions, we have revised the mentioned lines after including the information regarding single-parent family according to ABS report as follows:

“According to the Australian Bureau of Statistics (ABS) report (2013) on Australian families with children and adolescents, the majority of children (63–78%) in all age groups lived with both of their biological parents, 15-25% lived in a single-parent family, only 5–6% lived in a blended family, and 2–7% lived in a stepfamily.” [page:05; lines:109-112 ]

Comment 4: Lines 455-457: "The main disadvantage of this class was that no children and adolescents had come from intact families, and no one lived with either of their parents, thereby indicating relatively poor family functioning."

I suggest rephrasing the sentence as follows:

- The main disadvantage of this class was that no children and adolescents had come from intact families, and no one lived with both parents, thereby indicating relatively poor family functioning.

Authors response:

Thank you so much. We have rephrased this sentence as per your suggestions: 

“The main disadvantage of this class was that no children and adolescents had come from intact families, and no one lived with both parents, thereby indicating relatively poor family functioning.” [page:22 lines:458-460]

Comment 5: Lines 459-461: "In our study, we found that children and adolescents in classes 1 and 4 have very poor non-intact family structures and consequently possess weak family functioning."

I suggest rephrasing the sentence as follows:

- In our study, we found that most children and adolescents in classes 1 and 4 lived in non-intact families which is more often associated with poorer family functioning.

Authors’ responses: Thank you so much. We have rephrased this sentence as per your suggestions. [page:22-23; lines:462-464]

Comment 6: Could you please clarify this sentence? I don't understand what is meant here.

Lines 473-475: "The ABS report on Australian families with children and adolescents (2011) showed that the lowest percentage of adolescents 15–17 years old lived with both of their natural parents [13]."

Authors’ responses: Thank you for your comment.

Lines 473-475 meaning: According to ABS report, children under the age of five years were the most likely to live with both biological parents (78%), followed by those aged five to nine (72%). Those between the ages of 15 and 17 were the least likely to be in this situation (63%).

To emphasize the value of family structure, we brought this up (lines 473-475) in the discussion part in our study.

Comment 7: With regard to the conclusions, I suggest not to re-present the results again but to move the sentences from lines 482-490 into this chapter.

Suggestion for the conclusions chapter:

- "The present study has some important public health implications as it has identified the cluster of items that represent socio-demographic characteristics and revealed their associations with the mental and behavioral disorders of children and adolescents. The significant impact of individual class on the mental and behavioral disorders will help government and non-government agencies as well as practitioners differentiate the vulnerable classes from privileged classes in formulating health-related policy and strategy." The findings indicate that health promotion and prevention as well as combating poverty are needed to improve the mental health in particular among children and adolescents living in non-intact families and in families with a low socio-economic status. 

Authors’ responses:

Thank you for your suggestions. 

We have moved the mentioned lines from discussion part to conclusion part and revised the conclusion as follows: 

“The present study has some important public health implications as it has identified the cluster of items that represent socio-demographic characteristics and revealed their associations with the mental and behavioral disorders of children and adolescents. The significant impact of individual class on the mental and behavioral disorders will help government and non-government agencies as well as practitioners differentiate the vulnerable classes from privileged classes in formulating health-related policy and strategy. The findings indicate that health promotion and prevention as well as combating poverty are needed to improve mental health among children and adolescents living in non-intact families and in families with a low socio-economic status.” [page:25; lines:519-526]

---

## [Editor Report · Decision Letter 3]

5 May 2023

A latent class analysis of the socio-demographic factors and associations with mental and behavioral disorders among Australian children and adolescents

PONE-D-22-28764R3

Dear Dr. Afroz,

We’re pleased to inform you that your manuscript has been judged scientifically suitable for publication and will be formally accepted for publication once it meets all outstanding technical requirements.

Kind regards,

Petri Böckerman

Academic Editor

PLOS ONE
---

## [Editor Report · Acceptance letter]

10 May 2023

PONE-D-22-28764R3 

A latent class analysis of the socio-demographic factors and associations with mental and behavioral disorders among Australian children and adolescents 

Dear Dr. Afroz:

I'm pleased to inform you that your manuscript has been deemed suitable for publication in PLOS ONE. Congratulations! Your manuscript is now with our production department. 

Kind regards, 

on behalf of

Professor Petri Böckerman 

Academic Editor

PLOS ONE